# The interaction between endogenous GABA, functional connectivity, and behavioral flexibility is critically altered with advanced age

Kirstin-Friederike Heise [1,2✉], Laura Rueda-Delgado[1,3], Sima Chalavi[1,2], Bradley R. King[1,2,4], Thiago Santos Monteiro[1,2], Richard A. E. Edden[5,6], Dante Mantini [1,7] & Stephan P. Swinnen[1,2]

The flexible adjustment of ongoing behavior challenges the nervous system's dynamic control mechanisms and has shown to be specifically susceptible to age-related decline. Previous work links endogenous gamma-aminobutyric acid (GABA) with behavioral efficiency across perceptual and cognitive domains, with potentially the strongest impact on those behaviors that require a high level of dynamic control. Our analysis integrated behavior and modulation of interhemispheric phase-based connectivity during dynamic motor-state transitions with endogenous GABA concentration in adult human volunteers. We provide converging evidence for age-related differences in the preferred state of endogenous GABA concentration for more flexible behavior. We suggest that the increased interhemispheric connectivity observed in the older participants represents a compensatory neural mechanism caused by phase-entrainment in homotopic motor cortices. This mechanism appears to be most relevant in the presence of a less optimal tuning of the inhibitory tone as observed during healthy aging to uphold the required flexibility of behavioral action. Future work needs to validate the relevance of this interplay between neural connectivity and GABAergic inhibition for other domains of flexible human behavior.

[1] Department of Movement Sciences, Movement Control and Neuroplasticity Research Group, KU Leuven, Leuven, Belgium. [2] KU Leuven Brain Institute, Leuven, Belgium. [3] School of Psychology, Trinity College Dublin, Dublin 2, Ireland. [4] Department of Health & Kinesiology, College of Health, University of Utah, Salt Lake City, UT, USA. [5] The Russell H. Morgan Department of Radiology and Radiological Science, The Johns Hopkins University School of Medicine, Baltimore, MD, USA. [6] F. M. Kirby Research Center for Functional Brain Imaging, Kennedy Krieger Institute, Baltimore, MD, USA. [7] Brain Imaging and Neural Dynamics Research Group, IRCCS San Camillo Hospital, Venice, Italy. ✉email: kirstin.heise@kuleuven.be

Flexibly adjusting ongoing behavior poses a specific challenge to the neural control mechanisms and this becomes particularly visible with increasing age[1]. Functional deficits in endogenous γ-aminobutyric acid (GABA)-mediated neural signaling represent one suspect mechanism among the many potential causes of age-related behavioral decline[2]. GABAergic interneurons are suggested to have a major role in scaling and fine-tuning neural oscillations[3,4]. More specifically, GABAergic neurotransmission is believed to be an essential regulator of phase synchronization of neural oscillations[5], which has been proposed to constitute one of the brain's main modes of communication[6,7]. Thus, phase-based connectivity is indicative of the time-sensitive modulation of inter-site neural communication and therefore serves as a proxy for the responsiveness of the neural system.

An age-related decline of neural distinctiveness i.e., the recruitment of additional neural resources in the aging brain, has been associated with the age-related alteration of GABA concentration in perceptual and motor domains[8–10] and is meaningful to add to the original formulation of the dedifferentiation hypothesis of cognitive aging[11,12]. Consistent with this hypothesis, previous work indicates that GABAergic synaptic mechanisms on the cortical level, evaluated at resting state, predict the system's capacity for dynamic event-related modulation of cortical inhibition, and that this is linked to efficient motor control[13]. This work suggests that, once baseline GABAergic neurotransmission is imbalanced, the system's responsiveness is impaired, and this may have detrimental behavioral consequences. Such imbalance may occur at older age when disinhibition becomes more prominent. Experimental evidence for this association between age-related GABAergic dysfunction and declining behavior across perceptual and cognitive domains points towards a stronger impact on those types of behavior that require a high level of dynamic control (e.g.[14–16]). Yet, lowered motor cortical GABA levels are found to correlate with age-related changes in sensorimotor connectivity and diminished motor control[17]. These recent findings suggest a broader link between GABA availability and connectivity as a read-out for neural communication with implications for behavioral efficiency. However, whether these phenomena are simply co-occurring or whether they can be attributed to underlying causal mechanisms remains an open question.

Here, we chose a behavioral paradigm involving the dynamic control of transitions between dynamical motor states of varying complexity, which has shown to engage widespread, and in particular interhemispheric, neural communication within the sensorimotor system[18,19]. This behavioral paradigm is a prototype for flexible behavior, which involves a range of cognitive and motor control processes to perform these phase transitions[20]. By employing a multimodal approach to fuse endogenous GABA levels with the dynamic modulation of interhemispheric motor-cortical phase synchronization in the context of motor-state transitions in neurotypical young and older volunteers, we shed light on the nature of the interactions between task-related connectivity dynamics, behavior, and tuning of the motor-cortical inhibitory system during healthy aging.

## Results

To investigate the impact of individual variations in baseline GABA levels for the association between interhemispheric motor-cortical connectivity and complex bimanual behavior, we used a cross-sectional multimodal approach. The participants underwent in total three sessions, including magnetic resonance spectroscopy (MRS) in the first session and familiarization with the behavioral paradigm (motor-state transitions) in the second session. The third session followed 24 h after the familiarization and involved electroencephalography (EEG) during task performance.

MRS data were used to extract the endogenous GABA concentration. EEG data served to compute the task-related functional connectivity metric based on the circular variance of frequency-specific phase angle differences alongside the behavioral parameters (Fig. 1, see the Methods section for details). While the unimodal analyses (neurochemical, neural, behavioral) served to verify expected age differences, our primary interest was to integrate all three modalities to investigate the character of their interactions.

**GABA+ concentration.** To examine the endogenous motor-cortical GABA concentration, MRS data from left, right sensorimotor cortex (S/M1), and a control region, i.e., the occipital cortex (OCC) were acquired in 22 older and 22 young adults. In two cases (one older, one young), the data of the right S/M1 were excluded from further analysis due to motion artifacts and insufficient model fit. Consistency of the voxel placement across participants and individual traces of edited spectra for each voxel were visually inspected (Fig. 2). Quantitative quality metrics were comparable to those published in recent studies from our and other groups[21–23] (for descriptive statistics see Supplementary Table 1).

A gamma generalized linear mixed model (GLMM, identity link) was fitted to predict GABA+ with GROUP (young, older) and VOXEL (left S/M1, right S/M1, OCC) as factors of interest. All quality metrics (see "Methods" for details) and raw gray matter fraction (GM fraction) were added as covariates (after mean-centering) to identify their influence on GABA+ levels and their potential interaction with voxel or group through stepwise backward selection. This procedure revealed that of all quality metrics only GABA Fit error interacted with voxel and raw GM fraction interacted with group (Supplementary Table 2), all other interactions (all $p > 0.2$) were excluded from the final model. Of note, only interactions were removed during backward selection but all factors and covariates were kept in the final model to control for their influence. The final model (Supplementary Table 3) confirmed a significant effect of GABA signal-to-noise ratio (GABA SNR, Type II Wald $X^2(1) = 6.74$, $p < 0.01$) and Frequency offset (Type II Wald $X^2(1) = 17.20$, $p < 0.0001$). In addition, compared to the occipital voxel, both sensorimotor voxels tended to show higher GABA+ levels with increasing GABA Fit Error (VOXEL × GABA Fit Error (centered), Type II Wald $X^2(2) = 5.84$, $p = 0.05$). Relative to the young, the older adults showed overall lower GABA+ levels with increasing GM fraction ($\beta = -0.49 \pm 0.19$, 95%CI [−0.86, −0.12], $X^2 = -2.61$, $p < 0.01$, GROUP × raw GM fraction (centered), Type II Wald $X^2(1) = 6.82$, $p < 0.01$, Fig. 2d) across all voxels (for further discussion of this result see Supplementary Note 1).

With reference categories young and occipital voxel, we found an overall average GABA+ level around 2.86 i.u. (intercept $\beta = 2.86 \pm 0.26$, 95% CI [2.34, 3.37], $X^2 = 10.8$, $p < 0.0001$). Based on the Type II Wald statistics, GABA+ was found to be significantly different between age groups and this was specific to the voxel (GROUP × VOXEL $X^2(2) = 9.57$, $p < 0.01$, Fig. 2c). Specifically, marginal means contrast estimated for the individual parameter levels of the GROUP × VOXEL interaction revealed lower GABA+ levels in both sensorimotor voxels compared to the occipital voxel in the older (OCC-LEFT S/M1: $\Delta EMM = 1.65 \pm 0.227$, 95% CI [0.99, 2.32], $z = 7.29$, $p_{holm} < 0.0001$; OCC-RIGHT S/M1: $\Delta EMM = 1.564 \pm 0.2$, 95%CI [0.93, 2.20], $z = 7.27$, $p_{holm} < 0.0001$) while the young showed no differences between the voxels (marginal means contrasts given in Supplementary Table 4). Furthermore, the older showed significantly lower GABA+ levels in both sensorimotor voxels compared to the young (LEFT S/M1: $\Delta EMM = 0.64 \pm 0.15$, 95% CI [0.21, 1.07], $z = 4.40$, $p_{holm} < 0.0001$;

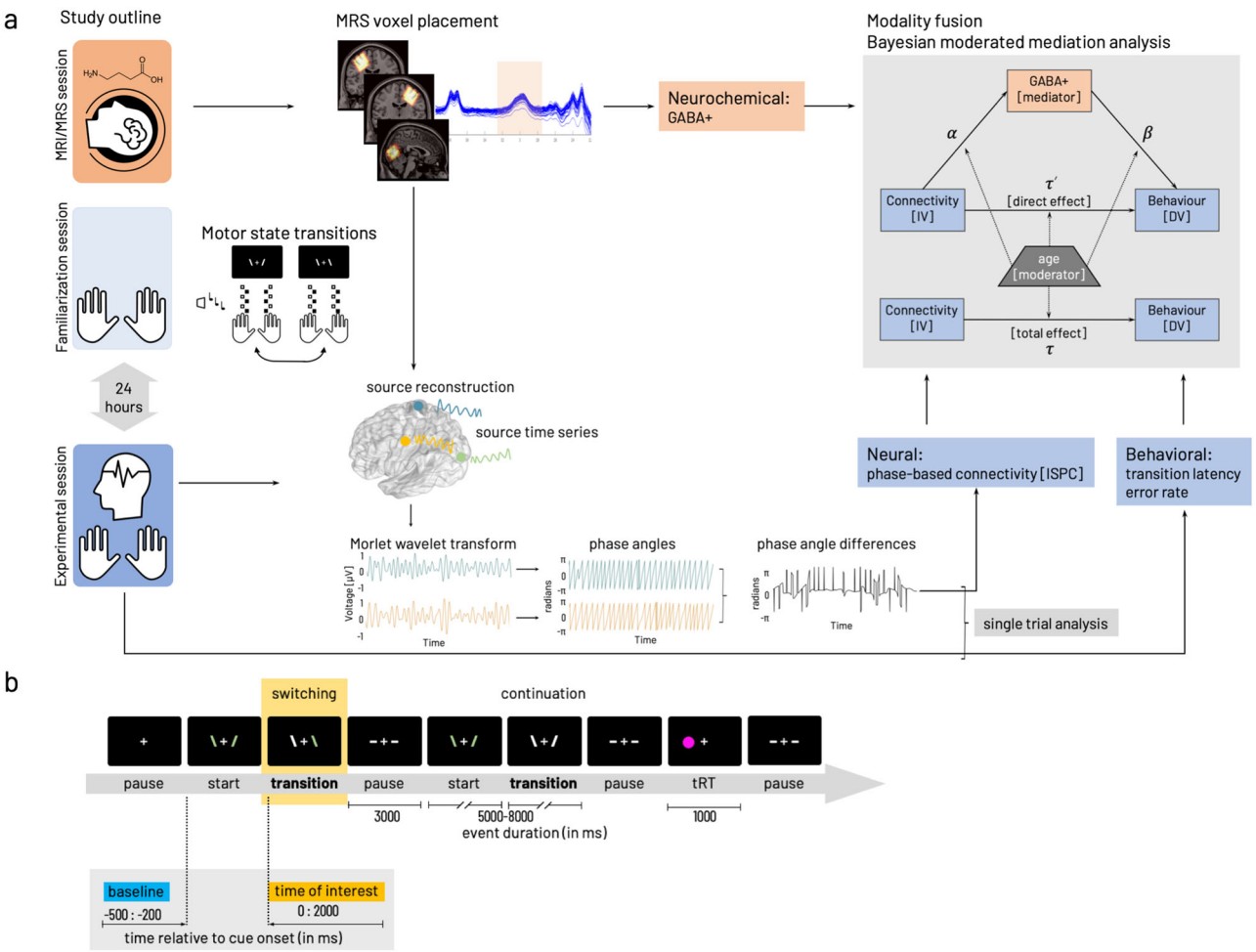

**Fig. 1 Experimental procedures and parameters of interest. a** Study outline with MRI/MRS (session 1), task familiarization (session 2) preceding the main experiment (session 3) including EEG during task performance. Edited MRS and T1-weighted images were used to extract tissue-corrected GABA levels and additional macromolecules (GABA+) from left and right sensorimotor and occipital voxels. The behavioral paradigm involved transitions between a stable (mirror-symmetric in-phase tapping of both index or middle fingers synchronously) and a less stable (anti-phase tapping, i.e., contralateral index and middle finger synchronously) motor state. Task familiarization included stimulus-response mapping and individual performance frequency adjustment. Performance in motor-state transitions was described with transition latency and error rate. The EEG signal was projected into source space based on the centroid coordinates of the GABA voxels. Phase angles were computed based on spectrally decomposed (Morlet wavelet transform) source time series. Phase angle differences between source signal pairs were used to compute connectivity (inter-site phase clustering, ISPC) between cortical sources. Phase angle differences were associated with behavioral performance in a single trial-based analysis. Then parameters of interest from the individual modalities (neurochemical, neural, behavioral) were integrated with a Bayesian moderated mediation analysis estimated for interhemispheric motor-cortical connectivity as independent variable [IV]. In both cases, dependent variable [DV] behavior was either median transition latency or cumulative error rate. Details on formalization of model paths $\alpha$, $\beta$, $\tau'$, $\tau$ given in "Methods". **b** Flow of events within the behavioral paradigm. Periods of finger movement (start, continuation, switching) were interleaved with rest periods (pause). A randomly occurring thumb reaction time task, a fast key press with either left or right thumb in response to appearance of a circle on the side of the required response was interspersed with the other events with a 5% probability of occurrence. Inlay highlights the time zones relevant for the analysis of behavioral data and EEG/EMG data analysis (time of interest, yellow). Data collected in the within-trial pause (demarked in blue) was used as baseline for the EEG/EMG analysis of the data from the time of interest (yellow). All parts of this figure have been created and published by the corresponding author on https://doi.org/10.6084/m9.figshare.19609842 licensed under CC BY 4.0.

RIGHT S/M1: $\Delta$EMM $= 0.55 \pm 0.12$, 95% CI [0.19, 0.91], $z = 4.48$, $p_{\text{holm}} < 0.0001$) but not the occipital voxel.

In short, controlling for quality metrics and raw gray matter fraction, we identified a relative reduction of GABA+ levels in the older compared to the young, which was specific for both sensorimotor voxels but not the occipital voxel.

**Behavior.** The control of transitions between motor states was tested with a variation of an established paradigm[24–27], in which the participants had to rhythmically tap in individually adjusted pace with the index and middle fingers of both hands and to control transitions between two coordinative patterns of different

complexity (in-phase/anti-phase, Fig. 1a). The behavioral data collected during the performance of the behavioral paradigm was analyzed in a time window of 2000 ms following the 'switching' cue. The time window of interest for these parameters was based on previous work[27] and pilot testing in older participants with the same task, which revealed that change in coordination mode is realized over an extended period. Furthermore, these previous results showed that a simple binarization of the precision (correct–wrong) does not reflect the ongoing adjustments made until the new coordination mode is mastered. Therefore, we aimed at quantifying performance with respect to (1) the precision (error rate) and (2) the speed (transition latency). Please see

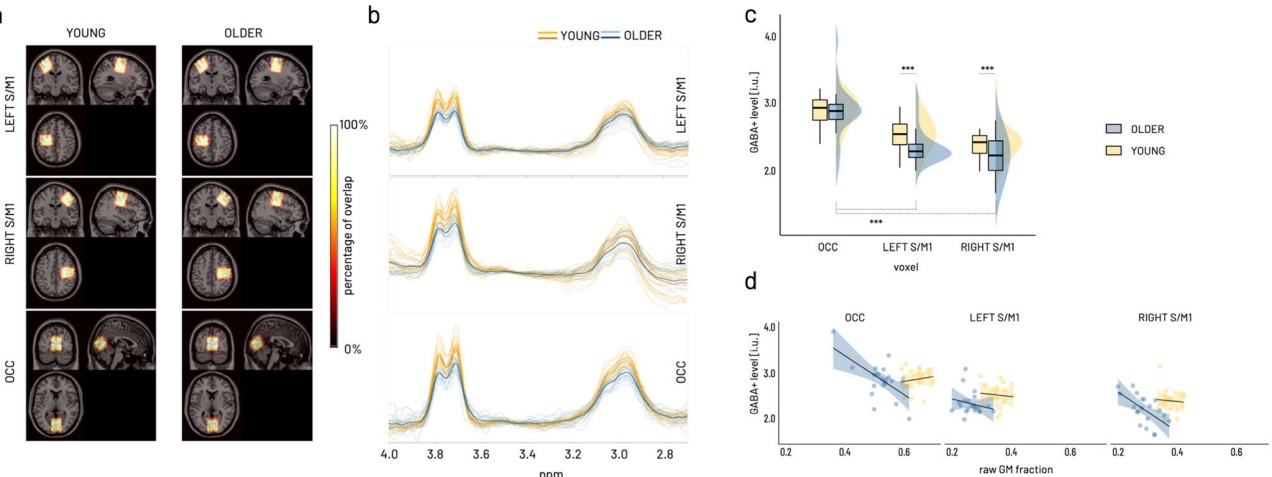

**Fig. 2 GABA+ spectroscopy results. a** Sum of individual GABA voxels projected into MNI space overlaid on standard brain template. Color coding indicates overlay agreement in percentage of all available images within group. Neurological display (i.e., coronal and axial view with left side on the left and right side on the right of image). **b** Individual edited spectra for LEFT S/M1, RIGHT S/M1, and OCC voxel color-coded for older (blue) and young (yellow) participants. Darker lines present average spectra per group (orange—young, dark blue—older). **c** Boxplots (lower/upper whiskers represent smallest/ largest observation greater than or equal to lower hinge ± 1.5 * inter-quartile range (IQR), lower/upper hinge reflects 25%/75% quantile, the lower edge of notch = median − 1.58 * IQR/ sqrt(n), middle of notch reflects group median) and distributions shown for the interaction effect of group and voxel on GABA+, which is driven by the differences between the occipital voxel and both sensorimotor voxels within the older in addition to the between age group differences for both sensorimotor voxels. Asterisks indicate significant effects of model-derived marginal mean contrasts corrected for multiple comparisons at ***$p_{holm}$ < 0.0001. **d** Age-group specific effect of raw gray matter (GM) fraction on GABA+ levels. Scatterplot (regression lines for subgroups with shading representing 95% CI) showing a relative decrease in GABA+ levels with increasing raw GM fraction in the older across all voxels ($p_{holm}$ < 0.0001). The analysis of GABA+ concentration included data from $N = 22$ older and $N = 22$ young participants for LEFT S/M1 and OCC voxels and from $N = 21$ older and N = 21 young for RIGHT S/M1.

Methods for details about the behavioral paradigm and parametrization of outcome parameters. On average $119 ± 20.5$ trials of individual transitions per participant were subjected to the analysis including $N = 21$ young and $N = 22$ older participants (descriptive statistics given in Supplementary Table 5).

*Error rate.* An overview of the distribution of error rate across age groups and transition modes is depicted in Fig. 3a. To capture a comprehensive picture of performance during the transition phase, we chose to split the precision measure into three distinct levels, namely, fully correct transitions representing transitions showing 100% correct tapping, failed transitions reflecting transitions with 100% of erroneous tapping, and cumulative error rate consisting of all remaining transitions not considered fully correct or failed.

*Failed transitions [trials with 100% error rate].* A logistic GLMM was used to predict failed transitions using group [OLDER, YOUNG], transition mode [in-phase, anti-phase], and number of trials as independent variables (full results in Supplementary Table 6). With around 0.3%, the overall odds of completely failing a transition were low (intercept for group = young, nTRIALc = 0, transition mode = IP: $\beta = −5.66 ± 0.60$ (odds ratio $0.003 ± 0.002$), 95% CI $[−6.83, −4.49]$, $X^2 = −9.475$, $p < 0.0001$). Based on the Type II Wald statistics, trial number significantly modulated the occurrence of failed transitions in a transition mode specific way and distinct for both age groups (GROUP × TRANSITION MODE × nTRIALc, $X^2(1) = 4.4$, $p = 0.04$, Fig. 3c left). Compared to the young, the older showed a higher number of failed trials early on and subsequently a steep decline of about 5% in likelihood of failed transitions from early to late trials for transitions into anti-phase (odds ratio = $−0.51 ± 0.17$, 95% CI $[0.27, 0.96]$, $X^2 = −2.09$, $p < 0.05$). Independent of group, transitions into anti-phase were twice as likely to fail than transitions into in-phase (odds ratio = $2.08 ± 0.54$, 95% CI $[1.24, 3.46]$, $X^2 = 2.80$,

$p < 0.01$, TRANSITION MODE Type II Wald $X^2(1) = 34.99$, $p < 0.0001$). Overall, with each additional trial, the odds of completely failing the transition tended to decline (odds ratio = $0.70 ± 0.14$, 95% CI $[0.47, 1.04]$, $X^2 = −1.78$, $p = 0.08$, nTRIALc Type II Wald $X^2(1) = 9.86$, $p < 0.01$) irrespective of group or transition mode.

*Fully correct transitions [trials with 0% error].* Like failed transitions, a logistic GLMM was fitted to predict fully correct transitions (full results in Supplementary Table 7). After removing failed transitions from the data, the overall odds for transitions to be fully correct were 4% (odds ratio = 0.038 for intercept: $\beta = −3.26 ± 0.28$, 95% CI $[−3.80, −2.72]$, $X^2 = −11.77$, $p < 0.0001$). Following the Type II Wald statistics, the two main explanatory parameters were GROUP ($X^2(1) = 15.43$, $p < 0.0001$) and TRANSITION MODE ($X^2(1) = 24.4$, $p < .0001$) and this was stable over number of trials. Remarkably, older participants were three times more likely to show completely correct trials (odds ratio = $3.52 ± 1.25$, 95% CI $[1.76, 7.045]$, $X^2 = 3.6$, $p < 0.001$), irrespective of transition mode. Compared to transitions into IP, switching into anti-phase was half as likely to result in fully correct transitions (odds ratio = $0.44 ± 0.10$, 95% CI $[0.29, 0.69]$, $X^2 = −3.63$, $p < 0.001$) independent of group.

*Cumulative error rate [0 < error rate/100 < 1].* A beta GLMM (logit link) was fitted to predict cumulative error rate including the same parameters as described above (full results in Supplementary Table 8). After excluding fully correct and fully erroneous transitions $[0 < error rate/100 < 1]$, transitions between transition modes in either direction involved around 20% of erroneous tapping, i.e., cumulative error rate (intercept: $\beta = 0.21 ± 0.04$, 95% CI $[0.15–0.29]$, $X^2 = −9.07$, $p < 0.0001$). Based on the Type II Wald statistics, the two main parameters influencing cumulative error rate were number of trials (nTRIALc, $X^2(1) = 6.692$, $p < 0.01$) and TRANSITION MODE ($X^2(1) = 4.91$, $p < 0.05$). Investigating the

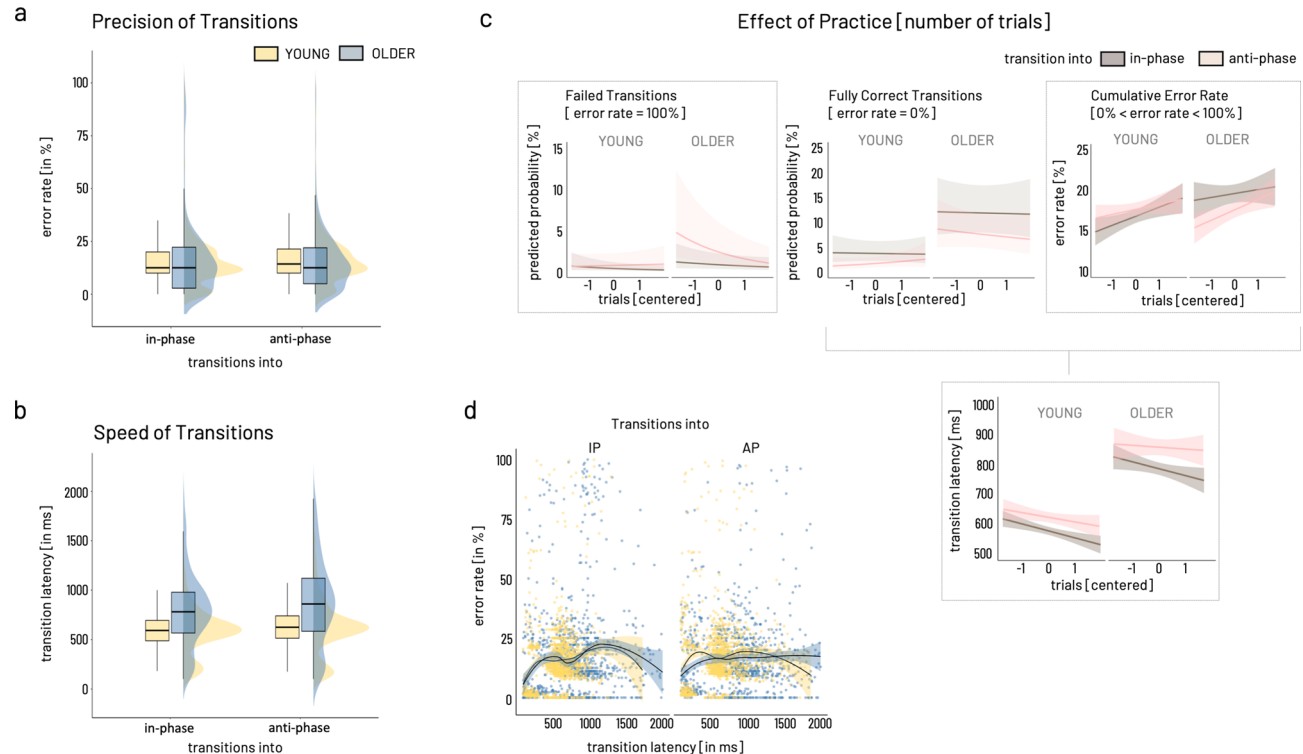

**Fig. 3 Predictors for the behavioral outcome. a** Error rate. Boxplots and distributions for overall error rate given separately for transition modes (in-phase, anti-phase) and age groups (blue: older, yellow: young). **b** Transition Latency. Color coding as in (**a**). **c** Effect of practice, i.e., number of trials (depicted as centered variable), are given for failed transitions, fully correct transitions, and cumulative error rate (from left to right), and transition latency (failed transitions excluded). Brown indicates transitions into in-phase mode, light pink depicting transitions into anti-phase. Frames around graphs indicate relevant modulation of the outcome over the number of trials, i.e., for failed transitions, cumulative error rate, and transition latency. Only in the case of failed transitions, older showed a significantly different modulation over time for transitions into anti-phase compared to the young with initially higher rate of trials with 100% error rate. Cumulative error rate showed a comparable increase across trials while transition latency decreased comparably in the two age groups and for both transition modes. **d** For both groups and transitions modes (into in-phase, into anti-phase), the relationship between speed and precision of transitions (excluding failed transitions) is non-linear as shown by locally weighted smoothing fitted over subgroups. Boxplots show lower/upper whiskers represent smallest/largest observation greater than or equal to lower hinge ± 1.5 * inter-quartile range (IQR), lower/upper hinge reflects 25%/75% quantile, the lower edge of notch = median − 1.58 * IQR/ sqrt(n), middle of notch reflects group median. Behavioral data analysis included $N = 22$ older and $N = 21$ young participants.

parameter estimates revealed that cumulative error increased about 7% over the number of trials irrespective of group or transition mode ($\beta = 1.07 \pm 0.02$, 95% CI [1.02, 1.12], $X^2 = 2.87$, $p < 0.01$). In comparison to transitions into IP, switching into anti-phase tended to yield around 6% higher cumulative error rate irrespective of group ($\beta = 1.06 \pm 0.03$, 95% CI [0.99–1.13], $X^2 = 1.78$, $p = 0.08$).

*Transition latency.* The transition latency was defined as the time delay between cue onset and valid response, i.e., the first occurrence of the correct transition mode indicated by the cue. Accordingly, failed transitions were excluded from the trials for the calculation of the transition latency. An overview of the distribution of transition latency across age groups and transition modes is depicted in Fig. 3b. A GLMM (Gamma family with a log link) was fitted to predict transition latency with the same independent variables described for error rate (full results in Supplementary Table 9). Given the model's reference categories, the average transition latency was estimated around 569 ms (intercept $\beta = 568.7 \pm 25.1$, 95% CI [521.68, 620.03], $X^2 = 143.98$, $p < 0.0001$). Based on the Type II Wald statistics, GROUP ($X^2(1) = 37.74$, $p < 0.0001$), TRANSITION MODE ($X^2(1) = 8.92$, $p < 0.01$), and number of trials (nTRIALc, $X^2(1) = 3.95$, $p < 0.05$) were the parameters explaining most of the transition latency's variance. The parameter estimates revealed, that older switched around 38% slower between transition modes compared

to the young ($\beta = 1.38 \pm 0.08$, 95% CI [1.22, 1.55], $X^2 = 5.22$, $p < 0.0001$). Transitions into the anti-phase pattern tended to be 7% slower than transitions into in-phase ($\beta = 1.07 \pm 0.04$, 95% CI [0.99, 1.17], $X^2 = 1.73$, $p = 0.08$). Independent of group or transition mode, transitions tended to become around 4% faster over time (nTRIALc, $\beta = 0.96 \pm 0.03$, 95% CI [0.91, 1.02], $X^2 = -1.37$, $p = 0.17$).

In summary, the behavioral results for error rate and transition latency show an expected slowing of the older participants but both age groups showed a decrease in transition latency across the experiment. However, the results show no general age-group effect on the precision of transition performance. While older seemed to have a slightly higher rate of failing transitions into the more difficult anti-phase mode early on, they showed an overall higher rate of completely correct transitions throughout the experiment compared to the young. The overall cumulative error rate, i.e., the percentage of erroneous taps in the course of a single transition, was comparable between the two age groups, showing an increase in errors as a function of practice (i.e., number of trials) and a trend of higher errors for transitions into the more challenging anti-phase mode. Additional support for comparable transition performance in both age groups comes from the results of the thumb reaction task (results in Supplementary Note 2, Supplementary Table 10), which neither show an effect of group nor interactions with transition mode or time across the experiment. However, it is necessary to

acknowledge that the thumb reaction time task poses additional cognitive load throughout the experiment and may, therefore, have interfered with the performance in mode transitions, particularly in the older participants. While we cannot rule out that this interference effect was present and that it has potentially affected the two age groups differently, we took a two-layered strategy to reduce this interference effect as much as possible (see "Methods section—Behavioral paradigm").

Finally, estimating the association between transition latency and error rate revealed a non-linear association of these two parameters for both age groups and transition modes (Fig. 3d). For both groups and transition modes, the speed-precision association may roughly be approximated with an inverted-U-shaped curve, potentially reflecting several underlying mechanisms beyond a linear speed-accuracy trade-off. Therefore, we argue that reducing the dimensionality of these two performance characteristics into one single measure appears not feasible. For subsequent analysis steps, trials of fully correct transitions and cumulative error rate were recombined into error rate, i.e., failed transitions were not considered for further analysis.

## Task-related modulation of phase-based connectivity (ISPC).
Phase-related connectivity (inter-site phase clustering, ISPC) between motor-cortical source signals was analyzed in $N = 20$ young and $N = 22$ older participants (see Methods for details about participant inclusion).

*Response-locked analysis of task- and group-related connectivity (ISPC) modulation.* No significant clusters were found for the interaction of group and transition mode. Subsequently, separate contrast analyses were performed to evaluate the effects of age group [YOUNG–OLDER] and transition mode [in-phase –anti-phase].

*Group contrast [YOUNG–OLDER].* A cluster showing a significant relative decrease of connectivity between the homologue S/M1 sources was evident for the mu to high beta frequency ranges (12–38 Hz) starting from −160 ms and lasting until 220 ms relative to the transition (Fig. 4a). This effect was driven by a strong reduction in connectivity in the young while the older showed increased connectivity overall but also when divided into separate time × frequency sub-clusters, reflecting pre-/post transition time zones and conventional frequency sub-bands. The sub-clusters spanned the ranges pre-transition high beta (−160–0 ms, >25 Hz), peri-transition low beta (−140–220 ms,15–25 Hz), and post-transition mu frequency range (>120 ms, 12–15 Hz, Fig. 4a, clusters A–C).

*Transition mode contrast [into in-phase–into anti-phase].* A single cluster was visible, extending mostly in the pre-transition time window in the mu to high beta range (Fig. 4b). Between −200 and 0 ms, a relative decrease in the full beta range (20–35 Hz) was evident (cluster A). This effect was driven by a decoupling before transitions into in-phase, while transitions into anti-phase were rather associated with an increase in left-right S/M1 connectivity before the actual transition was accomplished. Around the time of the transition, −50–60 ms, a relative increase in connectivity expanded over mu to beta range (cluster B), which was caused by an increased coupling for transitions into in-phase compared to transitions into anti-phase.

Taking the results of both contrasts together, interhemispheric motor-cortical connectivity showed clear age group differences in its spectral features during transitions. Furthermore, a modulation by transition mode (into in-phase versus into anti-phase) was also visible but we did not find evidence for altered connectivity in the older participants that was specific for transitions into one of the

two transition modes but rather a general change in connectivity pattern in the older adults.

## Single-trial phase angle difference—behavior association.
Our next interest was to further investigate the association between interhemispheric interactions and behavior. Because inter-site phase clustering (ISPC) is calculated across trials, no inference can be made about the intra-individual variations of the inter-site phase relationship and its association with variations in behavior. Linking inter-site interactions and behavior on a trial-by-trial basis allows interpreting the signature of this association and drawing conclusions about the behavioral relevance of the neural mechanisms. Therefore, frequency-specific phase angle differences between left and right S/M1 were extracted for each trial at the respective trial-based time of transition for the low (15–22 Hz) and high beta (25–30 Hz) frequency ranges identified in the respective time × frequency clusters during the previous analysis step (Fig. 4).

The distribution of phase angle differences between left and right S/M1 sources confirmed non-uniformity, i.e., significant clustering of phase angle differences around 0° for the young in the low beta range (15–22 Hz: $z = 70.43$, $p_{FDR} = 1.76e^{-30}$) and for both age groups in the high beta range around 0° for young and around 180° for the older (YOUNG: $z = 4.69$, $p_{FDR} = 0.03$, OLDER: $z = 7.26$, $p_{FDR} = 0.003$) when pooled over transition conditions (Supplementary Fig. 1).

To explore the role of the endogenous GABA+ concentration on this relationship, we dichotomized GABA+ concentration into below and above within group median concentration. Two-way ANOVA results showed that the factor group was a major source of variance for the average angle and that this was modulated by GABA+ level for both frequency ranges (15–22 Hz: GROUP $X^2(2) = 99.22$, $p < 0.0001$, GABA + $X^2(2) = 4.14$, $p = 0.13$, GROUP × GABA + $X^2(1) = 4.10$, $p = 0.04$; 25–30 Hz: GROUP $X^2(2) = 19.86$, $p = 4.9e^{-05}$, GABA + $X^2(2) = 8.75$, $p = 0.013$, GROUP × GABA + $X^2(1) = 9.99$, $p = 0.0016$ (see Fig. 5, additional results are given in Supplementary Note 3, Supplementary Table 11). Whereas, transition mode alone did not account for the variance in the data (15–22 Hz: TRANSITION MODE $X^2(2) = 0.57$, $p = 0.8$, GABA + $X^2(2) = 4.14$, $p = 0.13$, TRANSITION MODE × GABA + $X^2(1) = 5.38$, $p = 0.02$; 25–30 Hz: TRANSITION MODE $X^2(2) = 4.16$, $p = 0.13$, GABA + $X^2(2) = 8.75$, $p = 0.013$, TRANSITION MODE × GABA + $X^2(1) = 2.85$, $p = 0.09$). For both frequency ranges, circular-linear correlation revealed a significant association between phase-angle differences and quality of performance (i.e., error rate) following the transition. This association pattern was distinct for the two age groups in dependence of the relative (lower versus higher) GABA+ concentration. Specifically, when pooled over transition modes, phase angle differences were significantly associated with subsequent performance in the older adults in the low GABA+ subgroup. In the young, in contrast, a significant association was found for the high GABA + subgroup (15–22 Hz: OLDER$_{high\ GABA+}$ rho = 0.03, $p_{FDR} = 0.5$, YOUNG$_{high\ GABA+}$ rho = 0.08, $p_{FDR} = 5.45e^{-10}$; OLDER$_{low\ GABA+}$ rho = 0.07, $p_{FDR} = 5.45e^{-10}$, YOUNG$_{low\ GABA+}$ rho = 0.02, $p_{FDR} = 0.7$; 25–30 Hz: OLDER$_{high\ GABA+}$ rho = 0.03, $p_{FDR} = 0.9$, YOUNG$_{high\ GABA+}$ rho = 0.07, $p_{FDR} = 0.0005$; OLDER$_{low\ GABA+}$ rho = 0.06, $p_{FDR} = 0.007$, YOUNG$_{low\ GABA+}$ rho = 0.04, $p_{FDR} = 0.2$). Plotting error rate as a function of phase angle differences shows a variation in the trend of this association along the range from −360° to +360° phase lag, i.e., explaining the overall small correlation coefficient (Fig. 5c, d). Specifically, for both subgroups (older with lower GABA+, young with higher GABA+), a behavioral advantage for phase angle differences around 0° and

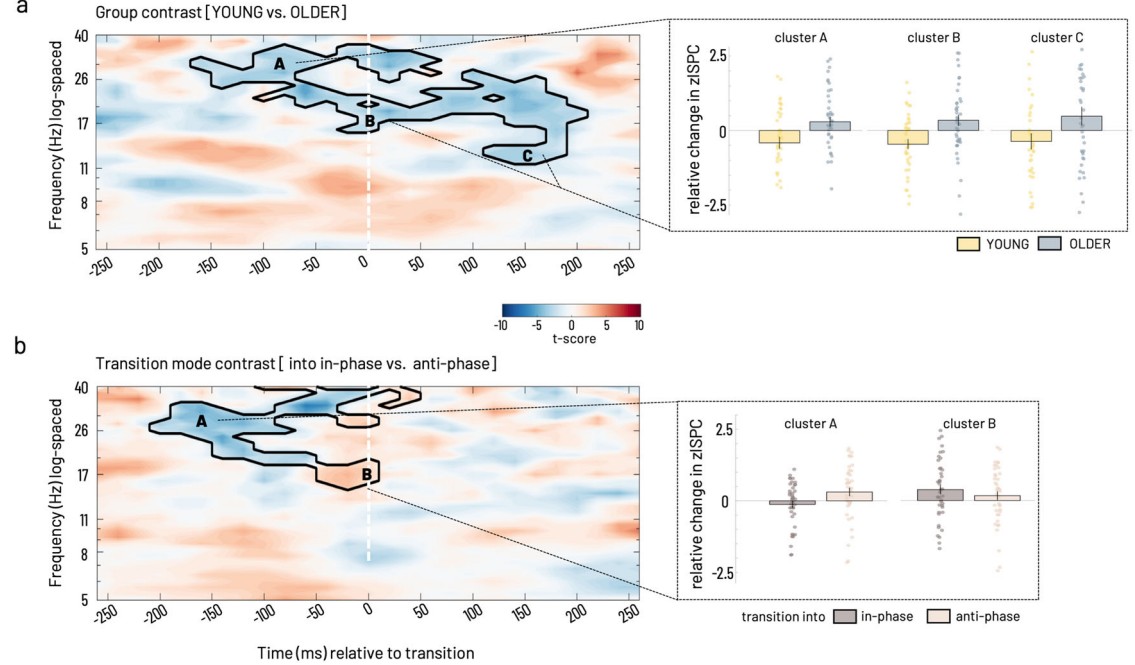

**Fig. 4 Statistical results of ISPC between left and right S/M1 source. a** For group contrast time-locked to the individual mean transition time. Cluster-corrected z maps for the test of GROUP contrast [YOUNG–OLDER, *t*-test against 0, *p* < 0.05, 2-tailed]. Color coding in the time-frequency resolved zISPC plot indicates *t*-values. Dashed vertical lines at 0 ms on the time axis indicate the individual median latency, i.e., the time of transition. **b** Statistical results for transition mode contrast time-locked to the individual mean transition time. Inlays show bar plots, which represent group averages (±SEM) of zISPC for respective cluster ranges indicated by capital letters for (**a**) group and (**b**) transition mode contrasts. Scatter plots depict individual participants' data for the group (yellow—young, blue—older) and transition mode (brown—in-phase, light rosé—anti-phase), respectively. Cluster-corrected z maps for the test of TRANSITION MODE contrasts [into in-phase–into anti-phase, *t*-test against 0, *p* < 0.05 2-tailed]. Bar plots show transition mode averages (±SEM) for respective cluster ranges. Phase-related connectivity (ISPC) data was analyzed in *N* = 22 older and *N* = 20 young participants.

higher subsequent error rate with phase angle differences of −180° and 180° were found.

To validate the specificity of the effects in terms of task-context and topography, the same analysis steps were run for two control conditions, namely the LEFT S/M1-RIGHT S/M1 phase lag at a random time point during baseline [start cue −300 ms], i.e., during between-trial pauses (Fig. 1b), and for phase angle differences for the OCC-L/RIGHT S/M1 connectivity at the time of transition. The analyses of the two control conditions revealed a significant GROUP x GABA+ modulation of the mean direction of phase angle differences between left and right motor cortical sources during the within-trial baseline [start cue −300 ms]. Furthermore, it showed a significant association between baseline phase lag and performance in the subsequent trial. Importantly, this association broadly resembled the pattern during transition described above although it was less specific for the within-age group GABA level in the low beta range (descriptive and inferential statistics in Supplementary Tables 12–14, Supplementary Fig. 2).

While we also found a significant GROUP x GABA+ modulation of the mean direction of the phase angle differences between the occipital source and both motor cortical sources at the time of transition for both frequency ranges, the pattern of the mean direction clustering was clearly different from that of the interhemispheric motor-cortical interaction in that it was not involving the clustering around 0° and 180°. Finally, no association between occipital–sensorimotor phase lag and behavioral performance was found (OCC-LEFT S/M1 and OCC-RIGHT S/M1 all $p_{FDR}$ > 0.1, descriptive and inferential statistics in Supplementary Tables 15–18, Supplementary Fig. 3).

In summary, single-trial phase angle differences at the time of transition showed to be different between the age groups, and this effect was modulated differently with the level of motor-cortical

GABA+ concentration. While in the young, the association between phase angle difference at time of transition and subsequent performance error was stronger under the relatively higher GABA+ concentration subgroup, the older showed a stronger association in the relatively lower GABA+ concentration subgroup. In both cases, 0° phase lag represented a behaviorally more advantageous state whereas a 180° phase lag was associated with more subsequent errors. This association was specific for the interaction between left and right sensorimotor sources and for the time of transition.

**Association between behavior and connectivity through GABA+.** To test the impact of baseline GABA+ levels on the relationship between interhemispheric motor-cortical connectivity and behavior in addition to the effect of age on the associations among all three variables (see Methods for details, schematic model structure given in Fig. 1a on the right), we employed a Bayesian moderated mediation analysis (including data from *N* = 22 older and *N* = 20 younger participants). For this purpose, we modeled the ISPC values extracted from the significant time × frequency sub-clusters of the response-locked analysis (independent variable), the median transition latency or error rate (dependent variable), the respective GABA+(mediator), and age (moderator) and estimated their associations in separate models for each of the individual connectivity pairs. The decision criterion for further investigation and discussion was a significant indirect (mediation) given the moderator age. Because all input variables were centered prior to modeling, it is necessary to keep in mind that conditional effects consequently need to be interpreted relative to the respective age group mean. As shown in the results below, for all significant models (see Fig. 6a for an overview), age was a relevant effect moderator of all model paths in the case of error rate and transition

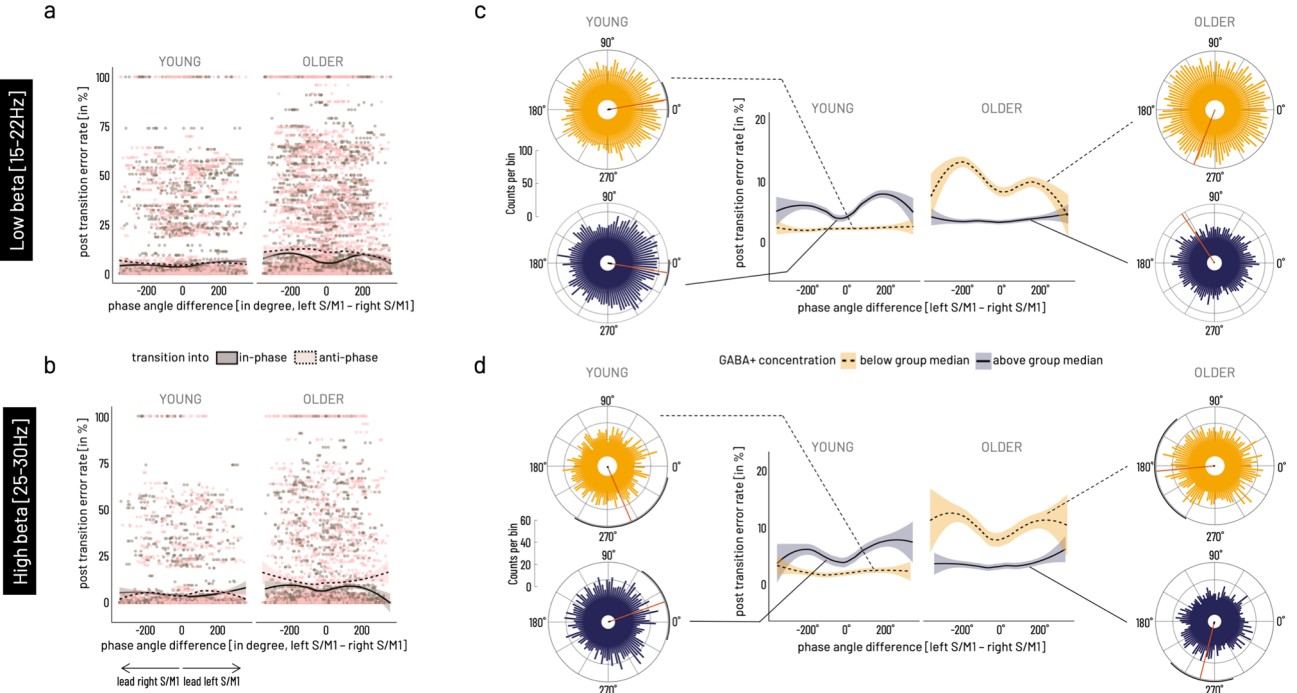

**Fig. 5 Association between cortico-cortical phase angle differences at the time of transition and subsequent performance error. a** Single-trial data shown for low beta [15–22 Hz] range. **b** Single-trial data shown for high beta range [25–30 Hz]. Data points represent single-trial data for transitions into in-phase (brown) and into anti-phase (light pink) mode, solid line indicates average phase angle difference—behavior association during transitions into in-phase mode, dashed line indicates average phase angle difference—behavior association for transitions into anti-phase mode. **c** Mean phase angle differences in the low beta frequency band were significantly modulated by factors age group and relatively higher versus lower GABA+ concentration when binarized into above group median (dark purple, solid lines represents subsample mean) versus below group median (yellow shading, dashed lines for subsample mean). **d** Mean phase angle differences in the high beta frequency band show a comparable pattern as in the low beta band. Rose plots show the histogram of binned phase angle differences with mean direction (red line) and 95% CI (black circumference) for significant non-uniformity of distribution. Phase angle differences for the low and high beta band were significantly associated with subsequent performance error in the young with relatively higher and in the older with relatively lower motor-cortical GABA+ concentration. In these subgroups, close to 0° phase lag was behaviorally beneficial (lower errors), while close to 180° phase lag was associated with higher performance errors. The analysis of single-trial phase-angle differences and performance error included data from $N = 22$ older and $N = 20$ young participants.

latency (Fig. 6b, c). Hence in the subsequent step, mediation results are shown conditional on the moderator age, highlighting predominantly opposing trends in the two age groups (regression coefficients for separate model paths given for all Bayesian moderated mediation models in Supplementary Table 19).

*Connectivity significantly predicts behavior in a time and frequency-specific manner and this relationship is moderated by age.* The main results for the models estimating the association between connectivity and behavior are graphically summarized in Fig. 6a for cumulative error rate and transition latency. Generally, models including right-hemispheric GABA+ levels yielded stronger evidence for mediation effects than those including left-hemispheric GABA+ for both age groups. Overall, the young group showed stronger evidence for mediation effects than the older. Stronger connectivity makes the young - but not the older - adults perform better. Simulating the total effect, i.e., the association between connectivity and behavior, conditional on moderator age reveals the opposing effects within the two groups (age group comparison depicted for one example model in Fig. 7a). For all three time-by-frequency clusters, the young participants show strong evidence that relatively stronger connectivity is associated with better performance, i.e., lower (i.e., relatively faster) transition time (pd = 100%) and lower error rate (pd = 100%, except pre-transition beta). In contrast, within the older adults, relatively stronger connectivity is associated with a relative slowing in transition latency (pd >

100%) but also with a trend for lower error rate (pd > 89%, except post-transition mu).

*Older adults benefit in precision (i.e., lower error rate) from higher non-dominant GABA+ levels.* For the association between connectivity in the peri-transition low beta band and error rate, both young (pd > 95%) and older (pd > 98%) show a negative indirect effect of right S/M1 GABA+ levels. This negative mediating effect has diverging consequences for behavior with respect to the two age groups (Fig. 7a). In the young, who showed better performance (i.e., lower error rate) with relatively stronger coupling, this direction of the connectivity-behavior association was more pronounced in the presence of lower non-dominant GABA+ concentration. In the older, who showed the opposite direction of the connectivity-behavior association (i.e., relatively higher error rates with stronger coupling), lower GABA+ concentration pronounced this direction. Higher right S/M1 GABA+ concentration, in contrast, ameliorated the association between stronger coupling and worse performance (i.e., higher error rate) in the older.

*Young adults are faster (i.e., shorter transition latency) with higher connectivity in the presence of lower non-dominant GABA+.* Simulating the mediation effect conditional on the moderator age, revealed a negative indirect effect of right hemispheric GABA+ (pd > 98–100%) on the association between connectivity on both behavioral outcomes, transition latency and, to a weaker extent

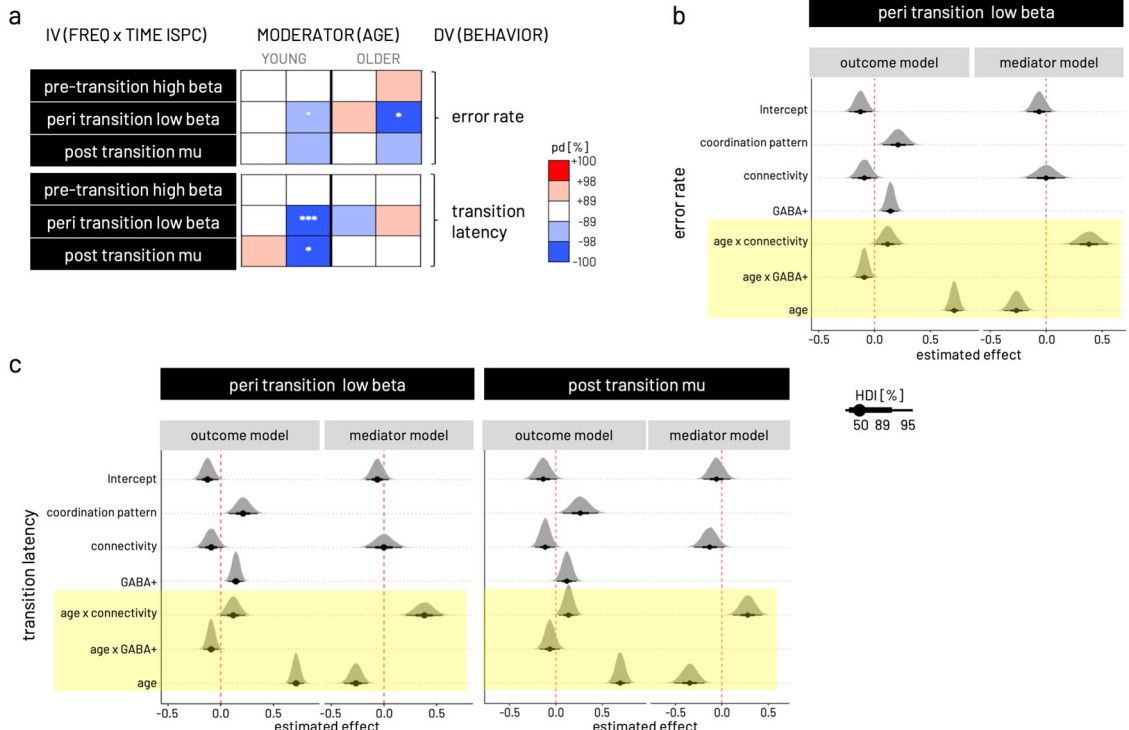

**Fig. 6 Results of Bayesian moderated mediation models. a** Overview over posterior directions (pd) for indirect (mediation) effects on error rate and transition latency conditional on upper/lower quintiles of moderator age (depicted as YOUNG and OLDER) for the models estimated with the independent variable (IV) based on the three time × frequency clusters derived in the response locked ISPC analysis. Models were run separately for left and right S/M1 GABA+ as mediator. The pd can be interpreted as the maximum probability of the effect accounting for the evidence derived from the data. Color coding of pd represents likelihood (in %) and direction of effect, i.e., red shading for positive effects and blue shading for negative effects. A pd of 95, 97.5, 99.5, and 99.95% corresponds to the frequentist two-sided *p*-value at the thresholds 0.1˚, 0.05*, 0.01**, 0.001*** respectively. See respective Methods section for further information about effect descriptors and measures of uncertainty used in Bayesian statistics. **b** Probability density plots for effects of parameters in the outcome and the mediator models input to the mediation analysis. Depicted are the three models with significant mediation shown in (**a**). The outcome model is shown for error rate and peri transition low beta ISPC. **c** Outcome models are shown for dependent variable transition latency and peri-transition low beta ISPC (left) as well as post-transition mu ISPC (right). For all three models shown in (**b**) and (**c**), the mediator model is related to right S/M1 GABA+. Highlighted in yellow are the significant effects of moderator age in all three models indicated by the posterior distributions of respective parameters falling with >99.5% on one side of the red dashed vertical line. Linewidth of black horizontal bars indicates 50, 89, and 95% highest density interval [HDI] of the parameters' effect. The Bayesian moderated mediation analysis was run based on data from N = 22 older and N = 20 younger participants.

(pd > 89–95%), on error rate in the young. Specifically, in the presence of lower right S/M1 GABA+ levels, the negative association between peri-transition low beta and transition latency was steeper in the young, i.e., in the presence of low GABA+ levels, stronger connectivity was associated with generally faster transitions while this effect was less pronounced in the presence of high GABA+ levels (Fig. 7b). This effect was comparable for connectivity in the high post-transition mu range and transition latency (Fig. 7c). The mediation effect between connectivity and transition latency was absent in the older for all time by frequency clusters (Fig. 7b, c). This absence of an indirect effect in the older can be explained by a weak association between the post-transition mu connectivity and GABA+ (path α) and in particular the absence of an association between right-hemispheric GABA+ and transition latency (path β) in both models.

In summary, the multimodal data fusion analysis revealed four main findings with respect to the potential mediating role of baseline GABA+ on the association between connectivity and behavior. First, baseline GABA+ levels exert an indirect effect on the link between interhemispheric motor-cortical connectivity in the low beta and high mu frequency band, time-locked to the behavioral event, and behavior. Second, variations in non-dominant hemispheric (right S/M1) GABA+ concentration was more likely to exert an indirect effect as compared to dominant

hemispheric sensorimotor GABA+ concentration. Third, individual variations in baseline GABA+ were found to exert an indirect effect in the young for models with speed and error rate, whereas an indirect effect for the older was only found in the model with error rate. Fourth, although the mediating effect of individual variations in baseline GABA+ is of the same direction in both age groups, it has diverging implications for the connectivity–behavior association in young and older adults. Importantly, the latter two points underline the overall expected finding of age being a strong effect moderator for mostly all bimodal relationships investigated here.

## Discussion
Flexibly adjusting ongoing behavior and switching between different modes of action is an essential ability in the human behavioral repertoire. Unfortunately, this flexibility declines across cognitive domains with increasing age[1]. Therefore, we tested dynamic motor-state transitions as a prototypical behavioral paradigm to investigate the effect of age on the interplay between endogenous GABA and the brain's responsiveness during flexible behavior. Based on this multimodal approach, we provide converging evidence for age-related differences in the preferred state of endogenous GABA+ concentration that is

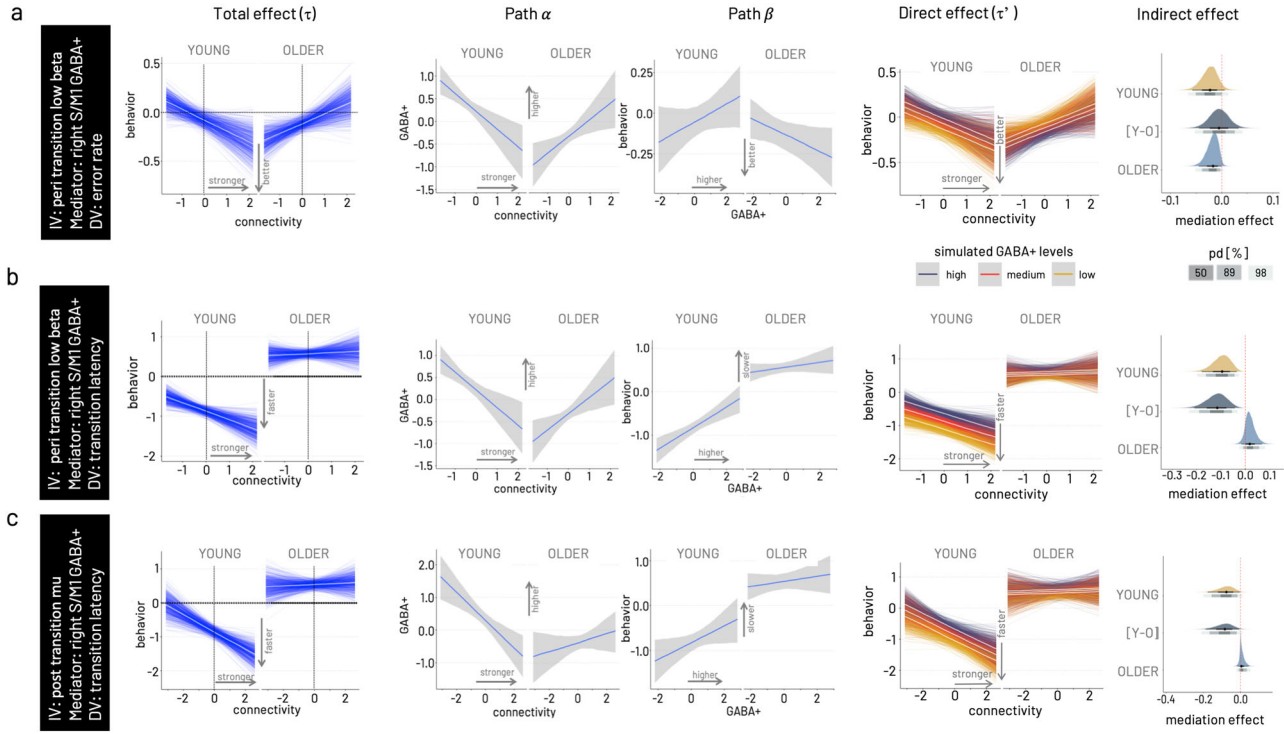

**Fig. 7 Results of Bayesian moderated mediation models.** From left to right, total effect (pathτ) conditional on moderator age, association between connectivity and GABA+ (path α), association between GABA+ and transition latency (path β), simulation of the mediation effect on the connectivity—behavior association (direct effect, path τ') for varying levels of GABA+ (low—yellow, medium—red, high—dark purple), and probability density ploy for the mediation effect conditional on age including the difference of young versus older [Y-O] mediation effects. In all three models, right hemispheric GABA + concentration is modeled as mediator. **a** Model for peri-transition low beta connectivity—error rate association. As visible in the total effect, the young behaviorally benefit from stronger S/M1-S/M1 connectivity in the low beta range, while the older show higher errors with stronger connectivity. The association between connectivity and GABA+ (path α) shows opposing trends in the two age groups, a negative association in the young and a positive association in the older. The association between GABA+ and transition latency (path β) is positive in the young and negative in the older. Simulations for the full moderated mediation model show that for the young the positive behavioral effect of stronger connectivity is more pronounced in the presence of relatively lower GABA+, while for the older the negative effect of increased connectivity is ameliorated in the presence of higher GABA+. Probability density plots of the mediation effect conditional on age for this model show a negative indirect effect for both age groups, and no difference between the age groups. Gray shading of probability of direction (pd) indicates limits of 50, 89, and 98% CI. **b** Model for peri-transition low beta connectivity—transition latency association. **c** Model for post-transition high mu connectivity—transition latency association. The Bayesian moderated mediation analysis was run based on data from $N = 22$ older and $N = 20$ younger participants.

optimal for interregional neural communication and that benefits flexible behavior. In conclusion, we tentatively suggest that the nature of the neural and neurochemical findings represents indicators for age-related compensatory mechanisms which serve to alleviate deterioration.

While the unimodal results highlighted the specific sensitivity of GABA+ concentration and phase-based interhemispheric motor-cortical connectivity to aging-related alterations, even in the absence of fundamental performance differences between the age groups, the evolving question became how the age-related changes may be expressed and reflect the underlying mechanisms of behavior.

Our unimodal results confirmed previous findings of the relevance of interhemispheric motor-cortical connectivity within the mu to beta frequency ranges for flexible behavior which underlies relevant age-effects[28]. Based on the observation that interhemispheric beta band decoupling was modulated by task complexity (i.e., more pronounced in the more complex transition mode), it may be interpreted as an indicator of inhibitory mechanisms necessary for the coordination of less congruent bimanual movements, as suggested by previous work[28–30].

Consequently, we linked inter-site interactions in the beta frequency range, detected in the unimodal analysis of connectivity at the time of transition, and behavior on a trial-by-trial basis. This analysis revealed an age-group-specific modulation of the interhemispheric phase lag between sensorimotor sources at the time of transition in addition to its association with the subsequent performance error. In an exploratory data-driven analysis, we found a first indication that phase lag at the time of transition and its association with subsequent behavior varied in dependence on the GABAergic state. Specifically, while in the young adults, relatively lower (than median GABA in the young) motor-cortical GABA+ concentration generally co-occurred with better performance irrespective of phase lag, relatively higher (than median GABA in the older) motor-cortical GABA+ concentration co-occurred with overall better performance in the older. In the respective less advantageous GABAergic state (i.e., relatively higher GABA+ in the young and relatively lower GABA+ in the older), 0° phase lag at the time of transition was followed by better performance, whereas a 180° phase lag was associated with more subsequent errors. These findings suggest a behavioral advantage through synchronization of interhemispheric sensorimotor sources with a phase lag of around 0°. Even though we may not rule out effects of volume conduction and source leakage on the inter-site phase relationship[31], we were able to confirm our results' temporal and regional specificity by comparing the interhemispheric motor-cortical phase lag during the transition with that at baseline and with the occipital-motor interaction. Previous work has suggested zero-phase lag for long-

distance connectivity in the beta frequency range between sensors covering left and right motor cortices during resting-state, as studied with magnetoencephalography[32]. Additional support for our findings' cogency comes from recent work that has proven the omnipresence of broadband zero-lag (i.e., 0° and 180° phase difference) functional connectivity, specifically for the homotopic brain regions based on intracranial recordings during varying vigilance levels in humans[33]. Although both studies have investigated spontaneous oscillations during resting-state, the authors speculated that functional connectivity around 0° phase lag might serve as a fundamental mechanism for the instantaneous integration of information from across brain regions allowing for predictive coding of expected events. Precisely, through long-distance synchronized oscillatory activity, the motor system might facilitate the anticipation of intrinsic or extrinsic cues allowing it to act with higher temporal precision. While our data support this hypothesis by showing a behavioral advantage of 0° phase lag at the time of transition, we also found an association between 0° phase lag during the within-trial baseline with performance of the subsequent trial, though less frequency-specific. During the within-trial baseline, the participants were required to remain attentive to the fixation cross and await the 'start cue'. Therefore, this observation suggests that the interhemispheric zero-lag synchronization might represent a more global state to potentially support the preparedness of the motor system. A mechanism to better anticipate the required behavioral action may have been specifically relevant in a less-well tuned system, i.e., less beneficial GABAergic state, and may represent a compensatory mechanism to uphold behavior.

To further investigate the indirect effect of GABA+ concentration on the relationship between phase-based connectivity and behavior, we used a Bayesian moderated mediation analysis integrating all three modalities. This analysis step confirmed on the one hand a steep age gradient for all bimodal interactions, i.e., all paths within the model, rendering the mediation analysis conditional on the moderator age highly meaningful. On the other hand, it revealed a hemispheric asymmetry of the mediator GABA+. Specifically, modeling the right hemispheric GABA+ concentration yielded higher evidence for an indirect effect on the connectivity-behavior relationship as compared to the GABA+ concentration of the left hemisphere. Given the non-directedness of the connectivity measure and the bimanual nature of the behavioral outcomes used here, the only variable differentiating hemispheric laterality is the mediator, i.e., GABA+ concentration. While we did not find a hemispheric difference in sensorimotor GABA+ concentration in either group in the unimodal analysis, the Bayesian model was sensitive to the actual variance. Previous MRS data from our own group support a hemispheric asymmetry in sensorimotor GABA+ concentration with lower concentration in the non-dominant hemisphere[34,35]. Electrophysiological data evidences an imbalance of phasic and tonic GABAergic inhibitory mechanisms within the motor system, also reflecting reduced fine-tuning of the non-dominant hemisphere across various age groups (e.g.[36-38]). Therefore, it is conceivable that the less well-tuned non-dominant hemisphere is more susceptible to excitation-inhibition variations and hence has a more pronounced effect on time-sensitive neural communication relevant for behavior, as suggested by our mediation results, irrespective of age. In addition to these two general findings, the mediation analysis delivered converging evidence for the two diverging states of beneficial GABA concentration in the two age groups as already suggested by the single-trial analysis of the phase angle differences.

We found an indirect effect of non-dominant GABA+ concentration on the connectivity-behavior association for both speed and precision in the young subgroup. In the presence of lower GABA+ levels, relatively stronger peri-transition beta band

and post-transition mu band coupling (i.e., less decoupling) was associated with better performance. This association weakened in the presence of higher GABA+ concentration in the young. Notably, the young showed overall higher GABA+ levels than the older for both motor cortex voxels. Hence, when interpreting the relative GABA+ concentration in the young, even lower levels are still comparably higher than the average seen in the older.

Computational modeling supports that extra-cellular GABA levels, most likely primarily detected with MRS[39,40], are critically influencing the variability in cortical neural activity and thereby define adequate information processing and integration[41]. Previous in vitro and in vivo work from animal models suggests that low extracellular GABA+ concentration represents the fine-tuned physiological environment with the optimal inhibitory tone for efficient and timely precise up- and downregulation of phasic synaptic inhibition (reviewed in ref. [42]). In support of these findings, experimental elevation of GABA+ concentration has shown to cause disturbances of neural processing, perception, and behavior in young healthy volunteers. Hereof, pharmacologically increasing endogenous GABA beyond physiological levels has shown to lead to exaggerated amplitudes of early evoked responses in somatosensory cortical areas[43] and decreased amplitudes of medium-latency evoked responses in the visual cortex[44]. Previous findings of elevated GABA concentration affecting both phasic and tonic inhibitory signaling of pyramidal and inter-neuronal cell populations in superficial and deep cortical layers may serve as a potential explanation[45–47]. While it is worth noting that lowering GABAergic concentration below physiological levels also has been shown to cause acute disturbance of spontaneous neural activity and perceptual processing in the primary visual cortex in young macaque monkeys[48,49], additional evidence for the detrimental functional effects of elevated GABA levels is available for sensorimotor processing. Strengthened movement-related desynchronization in the beta-frequency range detected over the primary motor cortex has been specifically linked to pharmacologically increased GABAergic drive[50]. In this former work, local desynchronization in sensorimotor beta-band oscillations, instead of peri-movement gamma-band or post-movement beta-band synchronization, was critically susceptible to pharmacological manipulation with benzodiazepines. In the present work, we found the indirect effect of GABA to be frequency-specific for response-locked modulation of long-distance synchronization in the mu and beta frequency range and its association with performance.

We therefore argue that the relatively lower endogenous GABA levels in the young reflects their neural system's preferred inhibitory state for effective neural communication, which assures the required responsiveness to modulate inhibition in the presence of dynamic task requirements.

In older adults, evidence for an indirect effect of GABA+ was restricted to the association between peri-transition beta-band connectivity and error rate. While the indirect effect of baseline GABA+ was negative as in the young, the implications for the connectivity-behavior association were the opposite as compared to the young. The detrimental effect of higher beta band connectivity on performance error in the older was ameliorated in the presence of higher GABA+ levels. In contrast to the young, relatively higher endogenous GABA appeared to represent the behaviorally more beneficial state in the older adults. This finding is, at first sight, intriguing, and the question is why the older do not benefit from the relatively lower GABA+ levels in the same way the younger adults do? However, retaining relatively higher GABA+ levels, i.e., closer to the concentration found in the young, probably reflects less age-related decline and subsequently lower impact on time-sensitive modulation of neural communication. Along these lines, higher GABA+ concentration has

been suggested to promote lower errors through optimal tuning of neural activity (reduced variability), promoting a better signal-to-noise ratio in the older. One effect of higher signal-to-noise is a more efficient perceptual filtering function from lower to higher level processing stages. A growing body of results from animal models (e.g.[48,49]), computational modeling[41,51], as well as results from aging human volunteers (e.g.[10]) supports this hypothesis. From this perspective, the mediating effect of GABA levels on the association between peri-transition beta-band connectivity and performance precision but not performance speed in older adults, as seen here, appears conceivable.

Previous work has shown reduced endogenous GABA+ levels to be linked to decreased resting-state network segregation, i.e., increased connectivity, and lower sensorimotor performance in older adults[17]. Although controversial findings exist, an increased interregional coupling has frequently been observed across imaging methods in older populations during task-free[52,53] and task-related conditions[54,55]. We observed a relative decoupling throughout the motor-state transitions for the interhemispheric mu to beta band connectivity in the young. In the older adults, in contrast, we found interhemispheric connectivity to be modulated on level of increased coupling during transitions. This finding of overall increased interhemispheric coupling provides support for the hypothesis of age-related dedifferentiation (e.g.[8,56]). Following the dedifferentiation hypothesis, it may be argued that the increased coupling reflects reduced processing efficiency in the older, which is potentially amplified by deficient inhibitory mechanisms as indicated by reduced GABA concentration.

The question as to whether alterations in GABAergic transmission reflect the cause or the 'cure' (i.e., compensation) for age-related neuronal functional decline reflected in behavioral performance deficits remains yet to be fully answered. For a complete picture it would be necessary to understand the task-specific modulation of inhibitory mechanisms, which requires the repeated or continuous evaluation of GABA+ concentration. A single resting-state measurement as in the present study offers only a limited insight into the functionality of GABAergic inhibition. In view of the limitations of the present work, it is necessary to point out that strictly speaking, a mediation implies the assumption of direct causality, which was not upheld in the present cross-sectional study. We, therefore, emphasize that our results do not allow to draw conclusions about causal mechanisms. Considering the collision or confound of many other factors modifying age-related changes of the brain-behavior interaction neglected here, our findings highlight the importance to investigate the nature of the interactions as a function of age. However, future work is needed to verify the generalizability of our findings for other aspects of flexible behavior in the motor and cognitive domain and other correlates of interregional communication. Finally, our Bayesian moderated mediation analysis, though hypothesis-driven, followed an exploratory approach and we acknowledge the lack of a cross-validation. Nonetheless, based on the converging evidence from our multimodal analyses, we conclude by proposing the increased interhemispheric connectivity to represent a compensatory mechanism, which is brought about by rhythmic entrainment of neural populations in homotopic motor cortices. Through this increased (potentially zero-lag) synchronization, the motor system is in a better state to anticipate and dynamically control motor action. This mechanism appears to be readily available in the young and healthy brain but seems to be most relevant in the presence of a less optimal tuning of the inhibitory tone to uphold the required dynamics of behavioral action as seen here in the older.

## Methods

### Ethics statement.
The protocol and all procedures of this study complied with the ethical requirements in accordance with the Declaration of Helsinki in its revised version from 2008, as approved by the Medical Ethical Committee of the KU Leuven (local protocol number S-58811, Belgian registration number B322201628182). All participants gave written informed consent to all of the study's experimental procedures and were reimbursed with 15 € per hour.

### Participants.
Forty-four volunteers (older group $N = 22$, age-range 62–82 years of age; young group $N = 22$, age-range 21–27 years of age) were recruited through local advertisements and were screened for in- and exclusion criteria. No statistical method was performed for an a priori sample size calculation; rather, we based reasoning for the selected sample size on numbers chosen in previous multimodal work (e.g.[17]). One young participant dropped out after the MRI data acquisition for personal reasons unrelated to the study. MRI, EEG, and behavioral data were thus collected in 21 young (10 women) and 22 older (11 women) participants. Due to technical problems, the EEG of one young participant had to be excluded, yielding different numbers of data sets included into the analysis for GABA, behavioral, and EEG analysis. All participants were right-handed, as evaluated with the Edinburgh Handedness Inventory[57] (laterality quotient: older 92.50 ± 0.20, young 85.00 ± 0.15, median ± 95% CI). All participants were free from neurological impairments and musculoskeletal diseases affecting the unconstrained movement of the fingers, did not take neuroactive drugs, and had normal or corrected-to-normal vision as evaluated with an in-house standardized questionnaire.

### Magnetic resonance imaging (MRI) and Magnetic resonance spectroscopy (MRS) acquisition.
MRS data acquisition and reporting was done following the Magnetic Resonance Spectroscopy quality assessment tool (MRS-Q)[58]. A 3D high-resolution T1-weighted structural image (repetition time = 9.5 ms; echo time = 4.6 ms; voxel size = $0.98 \times 0.98 \times 1.2$ mm$^3$; field of view = $250 \times 250 \times 222$ mm$^3$; 185 coronal slices) was acquired for each participant using a Philips Achieva 3.0 T MRI system and a 32-channel head coil. The $30 \times 30 \times 30$ mm$^3$ MRS voxels were positioned based on the T1-weighted image. For the left and right sensorimotor voxels, this was centered above the hand knob area[59] and rotated in the coronal and sagittal planes to align with the cortical surface of the brain. The occipital voxel was medially centered over the inter-hemispheric fissure, with the inferior boundary of the voxel aligned in parallel to the Tentorium cerebelli to cover left and right occipital lobes symmetrically[60].

Data were acquired using the Mescher–Garwood point resolved spectroscopy (MEGA-PRESS) sequence[61], with parameters resembling those of previous work[21–23]; 14 ms sinc-Gaussian editing pulses applied at an offset of 1.9 ppm in the ON experiment and 7.46 ppm in the OFF experiment, TR = 2000 ms, TE = 68 ms, 2000 Hz spectral bandwidth, MOIST water suppression, 320 averages, scan duration of 11 min, 12 s]. Sixteen water-unsuppressed averages were acquired from the same voxel. These scan parameters were identical for all three voxels.

MRS data were analyzed with the Gannet software 3.0 toolkit[62]. Individual frequency domain spectra were frequency- and phase-corrected using spectral registration[63] and filtered with a 3 Hz exponential line broadening. Individual ON and OFF spectra were averaged and subtracted, yielding an edited difference spectrum, which was modeled at 3ppm with a single Gaussian peak and a 5-parameter Gaussian model. The unsuppressed water signal serving as the reference compound[64], was fit with a Gaussian-Lorentzian model. The integrals of the modeled data were then used to quantify the uncorrected GABA levels. As discussed extensively, this method edits GABA as well as macromolecules at 3 ppm[65,66], therefore GABA levels reported are referred to as GABA+ (i.e., GABA+ macromolecules). To adjust GABA+ levels for heterogeneity in voxel tissue composition, MRS voxels co-registered to the high-resolution anatomical image were segmented into three different tissue classes, namely gray matter (GM), white matter (WM), and cerebrospinal fluid (CSF), with SPM 12 (http://www.fil.ion.ucl.ac.uk/spm/software/spm12/). The resulting voxel compositions were used to extract tissue-corrected GABA+ following the assumptions that GABA+ levels are negligible in CSF and twice as high in GM relative to WM[67], accounting for tissue-specific relaxation and water visibility values[67]. GABA+ levels were normalized to the average voxel composition within each age group after outlier removal[67]. Quality of the MRS data was assessed using the quantitative metrics GABA and the N-acetylaspartate signal-to-noise ratio (GABA SNR, NAA SNR), fit error of the GABA peak (GABA Fit Error), the drift (Drift) and the standard deviation of the water frequency offset (Frequency Offset), as well as linewidth, quantified as the full-width half-maximum of the modeled and N-acetylaspartate (NAA FWHM) signal.

### Behavioral paradigm.
The behavioral task involved two transition modes representing the two motor states, i.e., a mirror-symmetric synchronous tapping of homologue fingers (in-phase, the more stable motor state) and synchronous tapping of the index and middle finger of opposite hands (anti-phase, the less stable motor state). Since the anti-phase transition mode has been shown to represent the coordinatively more challenging pattern[18,20,24], tapping frequency was individually adjusted to 80% of the frequency with which the anti-phase pattern was comfortably performed without involuntary spontaneous transitions into the in-phase transition mode. This individual tapping frequency was auditory paced throughout the complete experiment. During the EEG session, the auditory pacing stimulus was provided through insert etymotic earphones with flat frequency response (Cortech Solutions, Wilmington, NC, USA). Tapping was performed on a

custom-made keyboard with six input keys (1000 Hz sampling rate). Visual target cues were presented on a standard 19" computer screen (refresh rate 60 Hz) and indicated which movement pattern to perform. Visual and auditory stimuli of the behavioral paradigm were programmed in LabVIEW 2016 (National Instruments, Austin/TX, USA). One complete trial consisted of a start cue subsequently followed by a cue to either continue with the same transition mode ('continuation') or transition into the respective other pattern ('switching' from in-phase to anti-phase, or vice versa, Fig. 1a). In this study, we focused on the switching transitions and thus the ratio of occurrence of continuation versus switching transitions was set to approximately 1:5 to yield enough trials for further analysis and keep participants from automatically switching. Trials were interleaved with pauses, which were always of the same length (3000 ms); the other events had a jittered inter-stimulus interval (5000–8000 ms). To preserve attention at a high level throughout the experiment, an additional thumb reaction time task (tRT) was included, which could occur instead of any other event type with a chance of 5%. The instruction was to respond as fast as possible upon cue occurrence (a magenta circle on left or right side of fixation cross) indicating either the left or right thumb to press the respective key. The tRT task was always followed by a pause with a latency of 1000 ms to avoid interference with transition performance. We followed a three-layered strategy to minimize the risk of interference of the tRT with the performance in the main task: First, we implemented a pause that directly followed each thumb reaction time task. This strategy was intended to reduce the effect on subsequent trials but does, of course, not preclude overall interference by the additional task. Second, we aimed at reducing the cognitive load of the tRT by increasing the salience of the imperative cues of the tRT (large magenta circles as imperative cues) and by employing different effectors than the main task (i.e., the thumbs instead of index and middle fingers). Finally, by implementing the stimulus-response matching through spatial congruency (i.e., left cue—left thumb, right cue—right thumb), we aimed at further reducing the cognitive load (e.g.[68]).

To minimize eye movements, participants were instructed to fixate a small cross in the center of the screen, which was visible at all times, during and in-between all cue presentations. For the within-trial pauses (described above) the instruction was to further attend to the fixation cross with minimal movement of the fingers or other body parts because these phases served as baseline for the EEG data analysis. Stimulus-response mapping was acquired during a training session held one day prior to the experiment. In this training session, a general familiarization with the keyboard was followed by the standardized frequency adjustment procedure. Subsequently, the visual cues were introduced with a visual presentation after which on average $44 \pm 21$ min (young: $36 \pm 16$ min., older: $51 \pm 23$ min.) of training were performed in the individual tapping frequency until the participants were able to successfully perform one block of 14 trials. In the main experiment, the individual tapping frequency was re-adjusted, and the participants performed in total 12 blocks of on average 14 trials each. Each block had a duration of approximately 4 min. Participants were given short breaks of individual length between each block to rest the eyes and make small movements.

**EEG recording and pre-processing.** Continuous EEG was recorded from 127 cephalic active surface electrodes (actiCAP, BrainProducts GmbH, Gilching/Germany) arranged according to the extended international. 10–20 system and referenced to the FCz electrode (implicit reference). Scalp-electrode impedance was kept below 20 kΩ. Data were acquired with a sampling rate of 1 kHz (BrainVision Recorder, version 1.21.0004, BrainProducts GmbH, Gilching/Germany).

Electrooculogram (EOG) was recorded using bipolar channels. For the EOG, silver/silver-chloride cup electrodes were placed on the left and right zygomatic processes (horizontal EOG) and on the left supraorbital process as well as on the sphenoid bone below the eye (vertical EOG).

All EEG data (pre-) processing and analyses were performed using functions from the EEGLAB toolbox version 2019.0[69], the Fieldtrip toolbox version 20190419[70], and customized Matlab functions (Matlab 2018b, MathWorks, Natick, MA, USA).

Off-line, data from EEG channels were high-pass filtered with a 1 Hz cut-off to remove baseline drift and down-sampled to 250 Hz. Line noise at 50 and 100 Hz was removed based on a frequency-domain (multi-taper) regression with the pop_cleanline function of EEGLAB. Subsequently, continuous data were segmented into epochs of 5 s length, ±2.5 s around the start cue (baseline) and the transition cue (time of interest) events to limit the effect of edge artifacts (Fig. 1b).

Thereafter, a rigorous artefact removal pipeline was employed to minimize the effect of high muscle-related artefact while ensuring sufficient data for subsequent analyses. This procedure included a combination of semi-automatic and visual inspection steps. First, bad channels were identified and removed (EEGLAB trimOutlier plugin with 2 µV as lower and 100 µV as higher cut-off for identification of bad channels). Then canonical correlation analysis (implemented in the EEGLAB AAR plugin)[71] was used to identify and remove excessive EMG activity present in the data due to the motor task (288 s window length and shift between correlative analysis windows, $10^6$ eigenratio, 15 Hz, ratio of 10, based on the welch algorithm). Thereafter, independent component analysis (runica/Infomax algorithm as implemented in EEGLAB) and SASICA was used as a semi-automatic procedure to inform removal of eye-movement -related and residual muscle artefacts[72]. For the identification of ICs

representing relevant artifacts, MARA, FASTER, and ADJUST algorithms were used, and components were rejected if they contributed ≥4% of the total data variance. Epochs with remaining muscle artefacts were removed based on trial-by-trial visual inspection. On average 50 trials per condition/participant went into further analysis. As a final step, available EEG channels were re-referenced to a common average reference.

**Localization of neuronal sources.** For the forward solution, an individual head model was created for each participant based on the same high-resolution structural MR image as used for the MRS analysis and 3D locations of the electrodes, registered with an optical infrared-camera based (NDI, Ontario, Canada) neuronavigation system (xensor™, ANT Neuro, Enschede, Netherlands). For the individual geometrical description of the head (mesh), the anatomical image was segmented into 12 tissue classes (skin, eyes, muscle, fat, spongy bone, compact bone, cortical gray matter, cerebellar gray matter, cortical white matter, cerebellar white matter, cerebrospinal fluid, brain stem), based on the MIDA model[73] using SPM12 (http://www.fil.ion.ucl.ac.uk/spm/software/spm12/)[74–76]. The EEG electrode positions were rigidly co-registered to the individual head surface (skin contour) by projecting the electrode coordinates in the native space through a rigid-body transformation, based on: (i) the estimation of anatomical landmarks (nasion, left/right peri-auricular points), (ii) the alignment of the electrode positions on the head surface through Iterative-Closest Point registration, and (iii) the projection of the electrodes onto the surface choosing the smallest Euclidean distance[77]. We chose conductivity values for each tissue class [in mS/m: cortical gray matter 333.3, cerebellar gray matter 256.4, cortical white matter 142.9, cerebellar white matter 109.9, brainstem 153.8, cerebrospinal fluid 1538.5, spongy bone 40.0, compact bone 6.30, muscle 100.0, fat 40.0, eyes: 500.0, skin 434.80] based on previous findings[78,79]. Dipole sources were constrained by a regularly spaced 6 mm three-dimensional grid spanning both the cortical/subcortical and the cerebellar gray matter. The volume conductor model was constructed based on a whole-head finite element model[80] using the SimBio toolbox (https://www.mrt.uni-jena.de/simbio) implemented in FieldTrip. To solve the inverse problem of describing the source activity, we made use of exact low-resolution brain electromagnetic tomography (eLoreta) algorithm[81], using a regularization factor $\lambda = 0.05$.

Source-space time-series were reconstructed using the pre-computed filter for three regions of interest, left and right sensorimotor, and occipital cortex. Coordinates for these regions of interest were extracted from the group (OLDER vs. YOUNG) averages of the individual centroid coordinates of the MRS voxels in MNI space and transformed into native space. We used a sphere with a 6 mm radius around the coordinates as a search grid to retrieve the gray matter grid voxel with the shortest distance to the coordinates of interest. Subsequently, singular value decomposition was used to reduce the dimensionality of the source activity time series in the target voxel from the x-, y- and z-components of the equivalent current dipole source to the projection that carried the maximal signal variance, i.e., the largest (temporal) eigenvector.

**Cortico-cortical connectivity.** To study the connectivity between cortical sources as a function of time, wavelet-based inter-site phase clustering, ISPC[82] was used. This phase-based connectivity measure depends on the distribution of the phase angle differences of two signals in polar space. The underlying assumption is that two neural sources are functionally coupled when their oscillations show temporal synchronization evidenced by angular differences. ISPC is a non-directional measure and has been shown to be less sensitive to time lags, non-stationarity of frequencies, and varying levels of noise[83].

To extract the phase angles, spectral decomposition was computed by convolving the ROI source signal with a set of complex Morlet wavelets, defined as complex sine waves tapered by a Gaussian[84]. The frequencies of the wavelets were chosen from 2 to 40 Hz in 50 logarithmically spaced steps to retrieve the full theta to the beta frequency range. The full-width half-maximum (FWHM) ranged from 400 to 104 ms with increasing wavelet peak frequency, corresponding to a spectral FWHM varying between 1.5 and 12 Hz[85]. Subsequently, ISPC was computed for 35 frequency steps from 5:40 Hz.

The phase angle differences were computed between ROI source signals over time and averaged over transition modes[82,86] on the down-sampled data (50 Hz) following

$$ISPC_f = \left| n^{-1} \sum_{t=1}^{n} e^{i\left(\phi_{xt} - \phi_{yt}\right)} \right| \qquad (1)$$

where $n$ is the number of time points, and $\phi_x$ and $\phi_y$ are phase angles from signals $x$ and $y$ at frequency $f$. Temporal modulation of ISPC change was evaluated in the time of interest (0 to +2000 post-cue, Fig. 1b) relative to the baseline period (−500 to −200ms) computed by subtracting the baseline ISPC values from the ISPC values in the time of interest.

In addition, instantaneous power was calculated by squaring the complex convolution results. Power spectra were normalized by converting the values to dB change relative to the fused within-trial baseline period, which was generated by averaging the time window between −500 and −200 ms before the cue over all start trials[87].

**Statistics and reproducibility**. The statistical analysis involved in a first step the analysis of the individual outcome modalities (MRS, behavior, and EEG) and in a second step the joined analysis of all three outcome modalities.

Generally, for all generalized linear mixed-effects models (GLMM) described hereafter, the goodness of fit was visually inspected based on the distribution of residuals. Models were fitted with a random intercept on subject level after validating that this improved model fit compared to the fixed-effects model. Model comparison was performed based on Akaike Information Criterion (AIC) and Bayesian Information Criterion (BIC). Parameter estimates for fixed effects and their interactions as well as 95% Confidence Intervals (CIs) and $p$-values were computed using Wald approximation. Parameter estimates for logistic models are reported as logits, i.e., log-odds, as well as odds ratios. In the case of the beta model with logit link, parameter estimates are reported as proportions and changes in rate of proportion. Standardized parameters were obtained by fitting the model on a standardized version of the dataset. Relevant interactions were followed up with contrasts for model estimated marginal means of parameter levels and reported as standardized differences ($\Delta$EMM $\pm$ standard error, 95% CI, $z$-value, $p$-value adjusted for multiple comparisons with Holm's method). Effect sizes are reported for the models' total explanatory power with conditional $R^2$ and for the fixed-effects part alone with marginal $R^2$ [88,89]. Forest plots are used to give an overview of the models' parameter estimates with CI, direction, and significance of their effects. Distribution and boxplots are used to represent summary statistics of group data. Computed variables for boxplots: lower/upper whiskers represent smallest/largest observation greater than or equal to lower hinge $\pm$ 1.5 * inter-quartile range (IQR), lower/upper hinge reflects 25%/75% quantile, the lower edge of notch = median − 1.58 * IQR/ sqrt(n), middle of notch reflects group median.

*Analysis of the MRS data*. GABA+ data were best estimated with a GLMM showing optimized fit modeling a gamma distribution and identity link function. Factors GROUP, VOXEL, and their interaction were modeled as fixed effects based on the study design variables. Random intercepts were fit on subject level. To identify the influence of the quality metrics and raw gray matter fraction (GM fraction) and their potential interaction with group or voxel, a stepwise backward selection approach was taken starting from a beyond optimal model with all covariates and their interaction with voxel or group. Based on the significance of parameters in the analysis of deviance (Type II Wald statistics), non-significant interactions were eliminated.

*Analysis of the behavioral data—error rate*. To analyze the occurrence of errors within the behavioral task, we chose to code and analyze three different aspects of the error information in the data to account for the skewed distribution of percentage data and the inherently zero-inflated data.

First, the data was transformed into a binary outcome, coding failed transitions, i.e., trials with an error rate of 100%. Binarization of the data was achieved by coding fully erroneous trials as "1" and all other trials as "0". This step was done based on the full set of available trials. A GLMM was used as a hurdle model and fit to the data with a Poisson distribution and logit link. Of note, trials with failed transitions were excluded from all subsequent analysis steps because our main interest was to investigate neural mechanisms underlying successful flexible transition behavior. Furthermore, the experimental paradigm yielded very low odds of completely failed trials, which did not allow to investigate mechanisms underlying failures in transition maneuvers.

Second, after removing the fully erroneous trials, the remaining data was transformed in a binary outcome coding fully correct transitions, i.e., an error rate of 0 coded as 1, versus erroneous trials, i.e., and error rate >0 coded as 0. As described in the first step, a GLMM was used as a hurdle model and fit to the data with a Poisson distribution and logit link.

Third, cumulative error rate in the trials not considered fully correct or fully erroneous, i.e., non-zero-inflated trials, were transformed into the range of the beta distribution [0 < error rate/100 < 1] and modeled as such using a GLMM with a logit link function. For all three error-rate-based outcomes, factors GROUP (old, young), TRANSITION MODE (into in-phase, into anti-phase), and covariate nTRIALc (trial number, centered) were entered into the model as fixed effects including all possible interactions. To account for intra-individual variability, random intercepts were modeled on subject level.

*Transition latency* did not follow a normal distribution and was therefore analyzed with a GLMM showing optimized fit assuming a gamma distribution and log link function. In analogy to the models for error rate, factors GROUP (old, young), TRANSITION MODE (into in-phase, into anti-phase), and covariate nTRIALc (trial number, centered) were modeled as fixed effects including all possible interactions. Random intercepts were modeled on subject level.

The association between transition latency and error rate (excluding failed transitions) was estimated for transition modes within age groups separately using a non-linear locally weighted smoothing fitted over subgroups.

*Thumb reaction time task (tRT)*. Thumb reaction time was computed as the response time latency (in ms) between the visual cue occurrence and the respective correct button press for the simple reaction time task. Thumb RTs were trimmed with a lower cut-off at 100 ms based on the assumption that a RT < 100 ms could unlikely be a true reaction to the visual cue and a high cut-off at within group mean

+3 SD. This conservative approach was chosen to retain as much data as possible without a priori assumptions regarding outlier features. Thumb reaction time was optimally fit with a gamma distribution and therefore a GLMM (Gamma family with identity link) was fitted to predict tRT with GROUP, SIDE, and nTRIALSc. Factors GROUP (older, young), SIDE (left, right), and covariate nTRIALSc (trial number, centered) were entered as fixed effects. Random intercepts were fit on subject level. Results of the thumb reaction time task are reported in Supplementary Note 2 and Supplementary Table 10.

*Analysis of the EEG data*. EEG data were analyzed with the focus on phase-related connectivity (ISPC) between motor-cortical sources. The statistical analysis of the task-related modulation of the spectral signature followed the pipeline described for ISPC and is outlined at the end of this paragraph. Additional results are presented in Supplementary Fig. 4 to allow the interpretation of the association/independence of ISPC and spectral power changes.

The effect of transition mode and age group on the frequency-band specific modulation of connectivity (inter-site phase clustering, ISPC) was analyzed in three steps. First, ISPC change from baseline was analyzed within-subject using a cluster corrected permutation (1000 permutations, two-tailed $t$-test, $p < 0.05$) to extract the effect size of change from baseline irrespective of transition mode. This step was used to extract the z-transformed ISPC changes (zISPC) per condition within-subject. In this and the subsequent steps, clusters were corrected for multiple comparisons and considered significant if they contained more time × frequency data points than expected under the null hypothesis at $p < 0.05$[90].

Second, group-level cluster-based permutation analysis (1000 permutations, two-tailed $t$-test, $p < 0.05$) of change in baseline-subtracted zISPC pooled over transition modes (stimulus-locked analysis) was used to confirm the relevance of connectivity modulation within the selected time and frequency windows. The results of this second step containing the stimulus-locked analysis of connectivity modulation are presented in Supplementary Fig. 5.

Third, in order to test the effect transition mode and its modulation by age group, differences of the z matrices were calculated for the transition mode contrast (in-phase – anti-phase) for the age groups separately and subsequently subtracted from each other ([in-phase – anti-phase]$_{YOUNG}$ – [in-phase – anti-phase]$_{OLDER}$). A two-sided $t$-test ($p < 0.05$) was then run with permuting the age group allocation (1000 permutations).

The third step was performed relative to the response, i.e., ±260 ms around the individual median transition latency specific for in-phase and anti-phase transitions, respectively (response-locked analysis). As task-related connectivity was not modulated by an interaction of condition and age group; therefore, both factors were tested subsequently in separate $t$-tests permuting the respective factor level allocation (1000 permutations, $p < 0.05$).

*Statistical analysis of task-related spectral power changes*. First, task-related power change, dB, from baseline were analyzed using a cluster corrected permutation (1000 permutations, two-tailed $t$-test, $p < 0.05$) within-subject to extract effect size of change from baseline irrespective of transition mode. This step was used to extract the z-transformed power changes per condition within-subject. Second, group-level cluster-based permutation analysis (1000 permutations, two-tailed $t$-test, $p < 0.05$) of change in power from baseline pooled over transition modes (stimulus-locked analysis) was used to confirm the relevance of spectral modulation within the selected time and frequency window. Third, in order to test the effect transition mode and its modulation by age group, differences of the z matrices were calculated for the transition mode contrast (in-phase – anti-phase) for the age groups separately and subsequently subtracted from each other ([in-phase - anti-phase]$_{YOUNG}$ – [in-phase - anti-phase]$_{OLDER}$). A two-sided $t$-test ($p < 0.05$) was then run with permuting the age group allocation (1000 permutations). The third step was performed relative to the response, i.e., ±260 ms around the individual median transition latency specific for transitions into in-phase and anti-phase, respectively (response-locked analysis). In all three steps, clusters were corrected for multiple comparisons and considered significant if they contained more time × frequency data points than expected under the null hypothesis at $p < 0.05$.

*Analysis of the association between phase-angle differences and behavior*. Frequency-specific phase angle differences between left and right S/M1 were extracted for each trial at the respective trial-based time of transition for the low (15–22 Hz) and high beta (25–30 Hz) frequency ranges identified in the respective time × frequency clusters during the previous analysis step. To rule out randomness of phase angle differences, non-uniformity of their distribution was tested using the Rayleigh test. A two-way ANOVA for circular data was used to test between-group differences and their interaction with GABA+ concentration. For this analysis step, artificial dichotomization of GABA+ concentration (into below and above within-group median concentration) was necessary[91]. Phase angle differences were then correlated with the single-trial error rate following the transition using circular-linear correlation. To validate the specificity of the effects in terms of task-context and topography, the same analyses steps were run for two control conditions, namely the LEFT S/M1-RIGHT S/M1 phase lag at a random time point during baseline [start cue −300 ms], i.e., during between trial pauses (Fig. 1b), and for phase angle differences for the OCC-L/RIGHT S/M1 connectivity at the time of

transition. All circular statistics and visual representations were performed with CircStat[92] and CircHist (https://github.com/zifredder/CircHist) Toolboxes implemented for Matlab 2018b and R package circular (version 0.4–93)[93]. All results are reported with FDR-corrected[94] $p$-values ($p_{FDR}$) to account for multiple comparisons across all subgroups.

*Analysis of the association between connectivity and behavior through GABA+.* The next goal was to get further insight into the relationship between EEG-derived connectivity metrics and behavior and the potential impact of endogenous GABA+ levels on this relationship in the presence of the effect of age. Therefore, we made use of Bayesian moderated mediation analysis modeling GABA+ as mediator and age as moderator variable and their impact on the relationship between cortico-cortical connectivity and behavior, i.e., transition latency and error rate (including cumulative error rate and fully correct transitions but excluding failed transition as described in the statistical analysis of the behavioral parameter error rate). This approach allowed us to further dissect the connectivity–behavior relationship given the individual variations of background GABA+ levels in the context of assumed aging-related changes of the associations between all variables. The Bayesian approach permits accounting for the non-gaussian data structure of the present sample and its size[95]. Conceptually, a moderated mediation model is built based on two regression models, in this case two generalized linear models, one that estimates the effect of the independent variables and relevant covariates (here the moderator) on the dependent variable (the outcome model, Eq. 2), and the second, which estimates the effect of the independent variable and relevant covariates on the mediator (the mediator model, Eq. 3):

$$Y = i_1 + c_1 X + c_2 W + c_3 XW + b_1 M + b_2 MW + e_1 \qquad (2)$$

$$M = i_2 + a_1 X + a_2 W + a_3 XW + e_2. \qquad (3)$$

In these models, $i_1$ and $i_2$ are intercepts, $Y$ is the dependent variable, $X$ is the independent variable, $M$ is the mediator, and $W$ is the moderator W interacting with each variable. In the outcome model (Eq. 2), $c_1$ is the coefficient relating the independent variable and the dependent variable, $b_1$ is the coefficient relating the moderator to the dependent variable, $c_2$ identifies the coefficient relating moderator and independent variable, the coefficients for the interactions with the moderator are $c_3$ and $b_2$. In the mediator model (E. 3.), $a_1$ is the coefficient relating the independent variable with the mediator, $a_2$ is the coefficient relating the moderator with the mediator, and $a_3$ is the coefficient for the interaction of the independent variable and the moderator. The residuals are identified by $e_1$ and $e_2$. These two models are combined within one multilevel model and estimated simultaneously for the moderated mediation analysis.

Here, a series of individual moderated mediation models was run for left S/M1-right S/M1 connectivity, using the respective zISPC pooled over the significant time × frequency clusters in addition to the GABA+ values of the corresponding voxel (e.g., model 1: predictor variable left S/M1 – right S/M1 zISPC, outcome variable transition latency, mediator variable left S/M1 GABA+, model 2: predictor variable left S/M1 – right S/M1 zISPC, outcome variable transition latency, mediator variable right S/M1 GABA+). Coordination pattern (in-phase – anti-phase) was included in the outcome model to account for its significant impact on both behavior and connectivity. Within each moderated mediation model, different associations (model paths) moderated by age were jointly estimated: (i) the association between independent and dependent variable in the absence of mediation (path τ, total effect); (ii) the association between independent variable and mediator (path α); (iii) the association between mediator and dependent variable (path β); (iii) the mediation effect (α*β, indirect effect); and (iv) the association between independent and dependent variable after adjusting for mediation (path τ', direct effect)[95]. A schematic of the moderated mediation model framework is given in the inlay in Fig. 1a on the top right.

All input variables were centered prior to fitting the GLMMs for outcome and mediator models using an exgaussian distribution, identity link functions (for mu, sigma, and beta), and uniform priors. Posterior distributions for multivariate models were obtained using Hamiltonian Monte-Carlo algorithm using Stan[96] implemented for R using brms[97,98] and rstanarm[99] packages. Four random walk chains each with 10.000 iterations discarding the first 1000 iterations (burn-in) were used for inference. Model convergence was examined using pareto-k diagnostics, approximate leave-one-out criterium (LOO), R-hat, and effective sample size (bulk-/tail-ESS); Bayesian $R^2$ served as an indicator for the quality of model fit. Median estimates and non-equi-tailed 89% credible intervals, i.e., Highest Density Intervals (89% HDI), are used to describe centrality and quantify uncertainties of the regression coefficients for the individual model paths accounting for their assumed skewness. To disentangle the influence of the mediator depending on variations of the moderator, a conditional process analysis was employed. Specifically, conditional estimates were simulated based on posterior draws versus highest sample quintiles of mediator and moderator. To allow for inferences about the relevance of the effects, probability of direction is reported (pd) for posterior probabilities, which can be interpreted as the probability (expressed in percentage) that a parameter (described by its posterior distribution) is strictly positive or negative when accounting for the evidence obtained from the actual data. The pd can take values between 50 (one half on each side) and 100 (fully on either side) and is approximated to a

frequentist two-sided $p$-value with the formula $p$-value $= 2*(1-pd/100)$[100,101]. Hence, a pd of 95, 97.5, 99.5, and 99.95% corresponds to $p$-value at the thresholds 0.1, 0.05, 0.01, 0.001.

**Reporting summary**. Further information on research design is available in the Nature Research Reporting Summary linked to this article.

## Data availability

Raw data are not publicly available due to European legal restrictions compromising the research participants' privacy and consent. All source data to reproduce Figs. 2–7 and related results are provided under https://doi.org/10.6084/m9.figshare.14256314.

## Code availability

Code to reproduce the figure, results and mediation models is provided together with the source data under https://doi.org/10.6084/m9.figshare.14256314. Code to reproduce the experimental set-up is available from the corresponding author [K.F.H.].

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

## Acknowledgements

We thank René Clerckx for software and technical support and Paul Meugens for technical support and skillful building of the technical equipment, and all volunteers for their motivation to participate. This work was supported by the Internal Research Fund KU Leuven (PDM/15/182) and the Research Foundation Flanders (1509816N, G089818N, G0F7616N, I005018N), the Excellence of Science grant (EOS, 30446199, MEMODYN), the KU Leuven Research Fund (C16/15/070), and Science Foundation Ireland (18/IF/6272).

## Author contributions

K.F.H. and S.P.S. designed the experiments. K.F.H. conducted the experiments. L.R.D., S.C., B.R.K., T.S.M., R.A.E.E., and D.M. contributed to acquisition and analytic tools. K.F.H. analyzed the data. K.F.H., L.R.D., S.C., B.R.K., T.S.M., R.A.E.E., D.M., S.P.S. contributed to the manuscript.

## Competing interests

The authors declare no competing interests.
