## [Peer Review File · Communications Biology]

Reviewers' comments:

Reviewer #1 (Remarks to the Author):

This manuscript describes a study in which ~20 younger and ~20 older adults underwent MRS to measure GABA in left M1, right M1, and primary visual cortex and also alternated between tapping patterns while EEG was recorded. The authors report slower responses and lower GABA+ levels in older adults in primary motor but not occipital cortex. Stronger interhemispheric connectivity was associated with better performance in young adults, but with a relative slowing in transition latency in older adults. Furthermore, phase angle differences were associated with performance in the older adults with lower GABA+ levels, while the association was stronger in the high GABA+ young adults. A mediation analysis found that baseline GABA+ levels exerted an indirect effect on the link between connectivity and behavior and this effect was stronger in right than left M1. The authors conclude that there are significant age-related differences in the relationship between GABA, connectivity, and behavior and suggest that increased connectivity in older adults may be compensatory.

The manuscript has a number of strengths. First, the methods and analysis struck me as sophisticated and rigorous. Second, the study incorporates a rare combination of EEG (to measure connectivity), MRS (to measure GABA), and motor behavior. Third, it provides evidence that the relationship between connectivity, GABA, and behavior is different in young vs. old.

On the other hand, I also thought the manuscript had some significant weaknesses. First and foremost, the work was not well motivated by a specific question, hypothesis or theoretical model. It therefore comes across as exploratory and it is difficult to relate the findings to existing theories in the cognitive neuroscience of aging. For example, do the results disconfirm any plausible theories in the literature? Second, the manuscript analyzed a large number of behavioral and EEG measures, many of the analysis choices seemed a little post hoc (e.g., dichotomizing subjects into low/high GABA, focusing on right hemisphere GABA in Fig 7), and there was no correction for multiple comparisons. This is particularly problematic given that the sample size was relatively small, no power analysis was provided, and there was no clear question/hypothesis going in. Third, I would like to see a comparable analysis of the left hemisphere GABA with behavior. If old and young still exhibit opposite patterns, then that would potentially undermine the compensation interpretation. Fourth, the work is purely correlational and does not admit causal inferences. Of course, that concern is not unique to this study, but applies to most neuroimaging work. Fifth, the manuscript only analyzes EEG data during transitions between two specific tapping patterns and it wasn't clear whether the results would generalize to other tasks and measures of connectivity.

Minor Things

- Line 28: has shown  has been shown
- Line 35: seen in the older adults  seen in older adults
- Line 46: to the neural control  to neural control
- Line 311: Taken the results  Taking the results
- Line 320: basis allows to  basis allows us to
- Line 382: transition showed to be  transition were
- Line 580: has shown to cause  has been shown to cause

Reviewer #2 (Remarks to the Author):

This study involves a complex experimental set up including MRS GABA measures, EEG beta connectivity measures and behavioural performance measures. It assesses the ability of two age groups (young and old) to switch between two different movement tasks, one in phase and one anti-phase. GABA is measured in both sensorimotor cortices and occipital lobe using MRS. Wavelet-based inter-site phase clustering (ISPC) is used for the EEG analysis. This is a novel and interesting combination of techniques and it is of great interest to see how GABA is involved in the interaction between oscillations and behavioural performance. The manuscript is well written, very dense and detailed and performs some complex analysis. I only have a few comments and suggestions as to improve it.

Abstract:

Could be written more clearly – I would advise splitting long sentences up into shorter sentences.

Materials and Methods:

I understand the reason for adding in the thumb reaction task in order to maintain vigilance, however do you think this would interfere with the main task as they might be constantly expecting the thumb task instead of focusing on the main task?

In the description of the EEG you say 10-10 system – do you mean 10-20 system as this is the most commonly used

EEG – recorded 127 electrodes. Say 10-10 system – do you mean 10-20 or did you use a 10-10 configuration?

In section "Analysis of the EEG data" you state that you analyse the ISPC between motor-cortical source and the signal from intrinsic hand muscles but everywhere else you simply state that you compare left and right sensorimotor cortex. Did you do any comparison with the hand muscles? If so this should be included somewhere, if it is a mistake then it needs to be removed and rewritten.

Results:

Figure 1 really helps to understand the details of this complex analysis

At some points in the manuscript you refer to primary motor cortex (both for MRS and EEG). With MRS and to a certain extent with EEG it is hard to state with confidence that you are only recording from M1 not S1 so I would alter it to sensorimotor cortex throughout.

I'm not an expert in mediation analysis so not sure if Bayesian moderated mediation analysis is most appropriate but it seems sensible from what you've described.

Figure 6 – what does pd stand for? You mention it represents likelihood but can you explain what pd stands for.

Discussion:

First paragraph – split sentence starting "using a multimodal approach..." into more, shorter sentences – hard to follow as it is and not concrete enough on findings – "GABA+ concentration to allow for" for example doesn't seem meaningful as it is – needs to be reworded a little to make clearer.

Reviewer #3 (Remarks to the Author):

This paper examines whether GABA mediates the link between functional connectivity and behaviour, and whether this mediation is moderated by age. This is an important contribution to the field and furthers our understanding of the age effects on associations between neural activity and behaviour. My main recommendation is to fine tune the writing a bit more so as to help the reader follow the arguments. Given that the methods are complex, it is easy for an unfamiliar reader to lose track of the various nuanced findings.

Major comments

1. Please discuss the significance of decreasing GABA with higher GM fraction in older adults. Along the same lines, what is the advantage of using tissue-corrected GABA values and also including GM fraction in the statistical models?
2. Why did you choose not to report the GABA concentration as a ratio to another compound like Creatine?
3. Single trial phase angle difference – behaviour difference: what does the association between phase angle difference and error rate look like when participants are not dichotomized based on GABA levels?
4. Supplementary fig.3 – please discuss this figure further. Especially in older participants, the pattern looks very similar to the findings about phase difference during the transition. While the findings are still interesting, if the same pattern is seen for baseline- and during transition- phase differences, interpretation of the data may need to be more nuanced.
I see that this issue is covered in the discussion – it would be helpful to edit the wording in the results section so it doesn't seem like this observation is being dismissed as insignificant.
5. Lines 457-461: the wording is unclear. Do both young and older adults have stronger associations between connectivity and behaviour in the presence of low GABA, but the associations are in different directions?

6. Please discuss the significance of the specific frequency bands that showed significant effects in the mediation analysis. For instance, in the context of previous literature, is there any reason why the low beta range is more important than high beta for this task?
7. Why did you choose to exclude failed transitions in the mediation analysis?
8. Please note that having a single GABA measurement, rather than task related GABA is a limitation.

Minor comments

1. Line 80 – please change the wording. At the moment, it sounds like there were 2 MRS sessions.
2. Line 166 – tap with both index and middle finger simultaneously?
3. Please make the sub-headings within the mediation analysis results more consistent. The analysis is complex and can be confusing for a reader who is not familiar with the data. For instance, in the sections on lines 454 and 464, it may be easier to use the terms error rate and latency in the headings instead of 'precision' and 'faster'.
4. Please check supplementary figure 1 - the a and b figures look identical, except the white dotted line. Also, the caption says 2a and b.
5. Line 783 – is it MIMA or MIDA?

Manuscript COMMSBIO-21-0749

Title: " The interaction between endogenous GABA, functional connectivity and behavioral flexibility is critically altered with advanced age"

We want to express our gratitude for the appreciation and the valuable comments of all three reviewers. We have carefully considered all their points and followed their suggestions as discussed point by point below. These revisions include additional presentation of data and results as well as major revisions of the text in the main manuscript and the supplementary material to improve clarity of our reasoning, methods, and interpretation of the results.

We feel that this comprehensive revision has improved the quality of our manuscript substantially.

Please find below our detailed responses to each of the reviewers' comments (identified with outside borders with our respective responses directly underneath) and the respective paragraphs highlighted (yellow) in the manuscript and supplements.

Reviewer #1 (Remarks to the Author)

This manuscript describes a study in which ~20 younger and ~20 older adults underwent MRS to measure GABA in left M1, right M1, and primary visual cortex and also alternated between tapping patterns while EEG was recorded. The authors report slower responses and lower GABA+ levels in older adults in primary motor but not occipital cortex. Stronger interhemispheric connectivity was associated with better performance in young adults, but with a relative slowing in transition latency in older adults. Furthermore, phase angle differences were associated with performance in the older adults with lower GABA+ levels, while the association was stronger in the high GABA+ young adults. A mediation analysis found that baseline GABA+ levels exerted an indirect effect on the link between connectivity and behavior and this effect was stronger in right than left M1. The authors conclude that there are significant age-related differences in the relationship between GABA, connectivity, and behavior and suggest that increased connectivity in older adults may be compensatory.

The manuscript has a number of strengths. First, the methods and analysis struck me as sophisticated and rigorous. Second, the study incorporates a rare combination of EEG (to measure connectivity), MRS (to measure GABA), and motor behavior. Third, it provides evidence that the relationship between connectivity, GABA, and behavior is different in young vs. old.

On the other hand, I also thought the manuscript had some significant weaknesses.

First and foremost, the work was not well motivated by a specific question, hypothesis or theoretical model. It therefore comes across as exploratory and it is difficult to relate the findings to existing theories in the cognitive neuroscience of aging. For example, do the results disconfirm any plausible theories in the literature?

We wish to thank the reviewer for the positive comments about our research, most notably highlighting the sophisticated, rigorous and multimodal methodology. We agree with the reviewer that the underlying theoretical framework for our study requires a more explicit introduction.

Following the reviewer's advice, we outlined the de-differentiation hypothesis of cognitive aging in the revised introduction:

“An age-related decline of neural distinctiveness i.e., the recruitment of additional neural resources in the aging brain, has been associated with the age-related alteration of GABA concentration in perceptual and motor domains⁸⁻¹⁰ and is meaningful to add to the original formulation of the dedifferentiation hypothesis of cognitive aging^{11,12}.”

[revised manuscript lines 55ff]

In our discussion, we refer back to this hypothesis and discuss the mediation results in light of dedifferentiation:

“We observed a relative decoupling throughout the motor-state transitions for the interhemispheric mu to beta band connectivity in the young. In the older adults, in contrast, we found interhemispheric connectivity to be modulated on level of increased coupling during transitions. This finding of overall increased interhemispheric coupling provides support for the hypothesis of age-related dedifferentiation (e.g.^{8,52}). Following the dedifferentiation hypothesis, it may be argued that the increased coupling reflects reduced processing efficiency in the older, which is potentially amplified by deficient inhibitory mechanisms as indicated by reduced GABA concentration.” [revised manuscript lines 640ff]

Second, the manuscript analyzed a large number of behavioral and EEG measures, many of the analysis choices seemed a little post hoc (e.g., dichotomizing subjects into low/high GABA, focusing on right hemisphere GABA in Fig 7), and there was no correction for multiple comparisons. This is particularly problematic given that the sample size was relatively small, no power analysis was provided, and there was no clear question/hypothesis going in.

The reviewer raises a very important point pertaining to multimodal studies with sample sizes like the one we present here. Considering the risks of analyzing complex relationships among several variables, we chose to account for the characteristics of the individual variables and their distributions for the unimodal analysis steps (see Material and Methods section - Statistical Analysis of MRS, Behavior, and EEG). Furthermore, for the integration of the different modalities we chose robust methods, such as the choice of procedures for the circular statistics and specifically the Bayesian Moderated Mediation Analysis. As discussed in the methods section, the Bayesian approach is specifically suited for the analysis of non-gaussian data structure and small sample size (revised manuscript lines 961ff). We agree that we need to provide further information about the difference in Bayesian inference compared to classical inferential statistics with respect to the need for ‘correction’. Classical hypothesis testing is based on the assumption that there is at least one statistically significant effect across a set of tests, which causes the inflation of Type I errors with each additional test. We

agree with the reviewer that multiple comparisons in classical inferential statistics are additionally of concern because in a setting where nonzero true effects do exist for some of the phenomena tested, multiple tests may identify additional statistically significant effects that do not exist. In Bayesian inference, in contrast, the basic assumption is that the null hypothesis is not believed to be strictly true. Within the Bayesian multilevel modeling framework, it is assumed that, when modelled correctly, reliable point estimates (e.g., median, mode) are retrieved with repeated testing. A Bayesian multilevel model “shifts point estimates and their corresponding intervals (e.g., Highest Density Intervals) toward each other (by a process often referred to as “shrinkage” or “partial pooling”), whereas classical procedures typically keep the point estimates stationary, adjusting for multiple comparisons by making the intervals wider (or, equivalently, adjusting the p values corresponding to intervals of fixed width)” (Gelman & (2012) *Journal of Research on Educational Effectiveness* 5, 189-211). As a consequence, Bayesian multilevel estimates are usually considered more conservative than classic inferential statistical estimates because they are more likely to include zero. Taken together, Bayesian multilevel models do not underlie the same assumptions as classic inferential statistics and, therefore, do not require correction for Type I errors, i.e. correction for multiple comparison.

We agree that we follow a data driven exploratory approach by dichotomizing the GABA+ data in the single-trial analysis of phase angle difference. We openly admit this in the methods section as well as in the presentation and discussion of results. In the revised manuscript, we further emphasized the exploratory nature of this analysis step:

“In an exploratory data-driven analysis, we found a first indication that phase lag at the time of transition and its association with subsequent behavior varied in dependence on the GABAergic state.” [revised manuscript lines 524ff]

Additionally, we now added the non-dichotomized results to the main manuscript and would like to refer to our answer to question 3 of reviewer 3 for their detailed discussion.

Third, I would like to see a comparable analysis of the left hemisphere GABA with behavior. If old and young still exhibit opposite patterns, then that would potentially undermine the compensation interpretation.

We agree that the description of results is not clear enough in explaining that this step of the analysis has been done and that results are shown in figure 6a. Additionally, we realized that the labelling of the overview in figure 6a is lacking important information that identifies the respective models. We have corrected the figure labelling now and added information in the accompanying text to better clarify that both left and right S/M1 GABA+ concentration have been modelled as mediators in separate models but only the models for the right S/M1 GABA+ showed higher evidence for a mediation effect conditional on moderator age (indicated with asterisks in figure 6a). Therefore, the

subsequent results are limited to these models. However, we report the regression coefficients (i.e., all model paths) for all mediation models in supplemental table 19 on which we based the subsequent conditional process analysis as described in the methods section [revised manuscript lines 1026ff].

The revised figure 6a was corrected and, in response to the reviewer's comment, we now added the information about modelling left and right hemispheric GABA+ as mediator in the legend:

Figure 6 Results of Bayesian moderated mediation models. a) Overview over posterior directions (pd) for indirect (mediation) effects on error rate (upper matrix) and transition latency (lower matrix) conditional on upper/lower quintiles of moderator age (depicted as YOUNG and OLDER) for the models estimated with the independent variable (IV) based on the three time \times frequency clusters derived in the response-locked ISPC analysis. Models were run separately for left and right S/M1 GABA+ as mediator.

[revised manuscript, figure 6a and legend]

We also revised the introductory text related to these results aimed at better informing the readers about the decision criteria involved in these analysis steps, which reads now as follows:

“To test the impact of baseline GABA+ levels on the relationship between interhemispheric motor-cortical connectivity and behavior in addition to the effect of age on the associations among all three variables (see Methods for details, schematic model structure given in Figure 1a on the right), we employed a Bayesian moderated mediation analysis. For this purpose, we modelled the ISPC values extracted from the significant time \times frequency sub-clusters of the response-locked analysis (independent variable), the median transition latency or error rate (dependent variable), the respective GABA+ (mediator), and age (moderator) and estimated their associations in separate models for each of the individual connectivity pairs. The decision criterion for further investigation and discussion was a significant indirect (mediation) given the moderator age. Because all input variables were centered prior to modelling, it is necessary to keep in mind that conditional effects consequently need to be interpreted relative to the respective age group mean. As shown in the results below, for all

significant models (see Figure 6a for an overview), age was a relevant effect moderator of all model paths in the case of error rate and transition latency (Figure 6b, c). Hence in the subsequent step, mediation results are shown conditional on the moderator age, highlighting predominantly opposing trends in the two age groups (regression coefficients for separate model paths given for all Bayesian moderated mediation models in Supplementary Table 19).”

[revised manuscript, lines 39ff]

Fourth, the work is purely correlational and does not admit causal inferences. Of course, that concern is not unique to this study, but applies to most neuroimaging work.

The reviewer raises of course an important point and we completely agree that claiming causal mechanisms is not justified based on our multimodal integration method of a cross-sectional data set. We discussed this limitation already in the previous version and further emphasized this limitation in the revised manuscript:

“In view of the limitations of the present work, it is necessary to point out that strictly speaking, a mediation implies the assumption of direct causality, which was not upheld in the present cross-sectional study. We, therefore, emphasize that our results do not allow to draw conclusions about causal mechanisms.” [revised manuscript lines 652ff]

Fifth, the manuscript only analyzes EEG data during transitions between two specific tapping patterns and it wasn't clear whether the results would generalize to other tasks and measures of connectivity.

We agree that the generalizability of our measures and results was not sufficiently addressed in the previous version of the manuscript. Therefore, we amended the introduction of the motor-state transition task to better highlight its relevance as a prototypical behavioral paradigm to test flexible behavior:

“Here, we chose a behavioral paradigm involving the dynamic control of transitions between dynamical motor states of varying complexity, which has shown to engage widespread, and in particular interhemispheric, neural communication within the sensorimotor system^{18,19}. This behavioral paradigm is a prototype for flexible behavior, which involves a range of cognitive and motor control processes to perform these phase transitions²⁰.”

[revised manuscript lines 72ff]

Furthermore, we added a discussion of the limitation of our findings and the need to critically assess their generalizability with respect to behavior and measures of connectivity:

“However, future work is needed to verify the generalizability of our findings for other aspects of flexible behavior in the motor and cognitive domain and other correlates of interregional communication.” [revised manuscript lines 658ff]

Minor Things

Line 28: has shown  has been shown
 Line 35: seen in the older adults  seen in older adults
 Line 46: to the neural control  to neural control
 Line 311: Taken the results  Taking the results
 Line 320: basis allows to  basis allows us to
 Line 382: transition showed to be  transition were
 Line 580: has shown to cause  has been shown to cause

We proof-read the indicated sections carefully and revised them as summarized below:

Based on the recommendation of the reviewer, we revised the abstract including lines 28 and 35.

“Taking the results of both contrasts together, interhemispheric motor-cortical connectivity showed clear age group differences in its spectral features during transitions.”

[revised manuscript, line 309f]

“Linking inter-site interactions and behavior on a trial-by-trial basis allows interpreting the signature of this association and drawing conclusions about the behavioral relevance of the neural mechanisms.” [revised manuscript, line 318ff]

“In summary, single-trial phase angle differences at the time of transition showed to be different between the age groups, and this effect was modulated differently with the level of motor-cortical GABA+ concentration.” [revised manuscript, line 386ff]

“While it is worth noting that lowering GABAergic concentration below physiological levels also has been shown to cause acute disturbance of spontaneous neural activity and perceptual processing in the primary visual cortex in young macaque monkeys^{39,40}, additional evidence for the detrimental functional effects of elevated GABA levels is available for sensorimotor processing.” [revised manuscript, line 601ff]

Reviewer #2 (Remarks to the Author):

This study involves a complex experimental set up including MRS GABA measures, EEG beta connectivity measures and behavioural performance measures. It assesses the ability of two age groups (young and old) to switch between two different movement tasks, one in phase and one anti-phase. GABA is measured in both sensorimotor cortices and occipital lobe using MRS. Wavelet-based inter-site phase clustering (ISPC) is used for the EEG analysis. This is a novel and interesting combination of techniques and it is of great interest to see how GABA is involved in the interaction between oscillations and behavioural performance. The manuscript is well written, very dense and detailed and performs some complex analysis. I only have a few comments and suggestions as to improve it.

Abstract:

Could be written more clearly – I would advise splitting long sentences up into shorter sentences.

We thank the reviewer for the overarching positive comments about our research. Following the reviewer's suggestion, we revised the abstract to improve readability. It now reads as follows:

“The flexible adjustment of ongoing behavior challenges the nervous system's dynamic control mechanisms and has shown to be specifically susceptible to age-related decline. Previous work links endogenous gamma-aminobutyric acid (GABA) with behavioral efficiency across perceptual and cognitive domains, with potentially the strongest impact on those behaviors that require a high level of dynamic control. Our analysis integrated behavior and modulation of interhemispheric phase-based connectivity during dynamic motor-state transitions with endogenous GABA concentration. We provide converging evidence for age-related differences in the sweet spot of endogenous GABA concentration for more flexible behavior. We suggest that the increased interhemispheric connectivity observed in the older participants represents a compensatory neural mechanism caused by phase-entrainment in homotopic motor cortices. This mechanism appears to be most relevant in the presence of a less optimal tuning of the inhibitory tone to uphold the required flexibility of behavioral action.”

[revised manuscript Abstract]

Materials and Methods:

I understand the reason for adding in the thumb reaction task in order to maintain vigilance, however do you think this would interfere with the main task as they might be constantly expecting the thumb task instead of focusing on the main task?

We agree that this is indeed a valid point. We are aware of the possibility that an additional task would potentially increase the cognitive load and could cause interference with the main switching task. In the planning of this study, we also considered that this cognitive interference is stronger with increasing age. The problem with separate tasks to evaluate the level of attention such as the Psychomotor Vigilance Task [Dinges & Powell (1985). Behavior Research Methods, Instruments, & Computers, 17(6), 652–655 <https://doi.org/10.3758/BF03200977>] is that they would have added additional time to the already lengthy experimental set-up. Furthermore, in order to monitor sustained attention during the main task, we decided that this evaluation had to be interleaved with the main task and not as a separate experiment. In order to reduce the interference effect of the thumb reaction time task on the performance in the motor-state transitions (the main paradigm), we implemented a pause that directly followed each thumb reaction time task (described in lines 750ff). This strategy was intended to reduce the effect on subsequent trials but does, of course, not preclude overall interference by the additional task. To limit overall interference by the additional thumb-reaction time task as much as possible, our attempt was to keep the cognitive load of the attention task as low as

possible. Therefore, the task was designed to be very salient (large magenta circles as imperative cues) and use different effectors than the main task (i.e., the thumbs instead of index and middle fingers). Although being a choice reaction time task, by implementing the stimulus-response matching through spatial congruency (i.e., left cue – left thumb, right cue – right thumb), we aimed at further reducing the cognitive load (see e.g., Hommel (1993) *Psychological Research*, 55(4), 270–279 <https://doi.org/10.1007/BF00419687>). We agree that the potential interference and the likelihood of a more critical effect in the older presents a limitation of the study design and therefore add an additional discussion of this point. Considering the amount of information discussed in the main manuscript, we suggest to add this as supplemental discussion provided together with the methods and results of the thumb reaction time task:

“Limitations of the Thumb reaction time task

It is necessary to acknowledge that the tRT poses additional cognitive load throughout the experiment and may, therefore, might have interfered with the performance in mode transitions, and in the older participants in particular. While we cannot rule out that this interference effect was present and that it has potentially affected the two age groups differently, we took a two-layered strategy to reduce this interference effect as much as possible. First, we implemented a pause that directly followed each thumb reaction time task. This strategy was intended to reduce the effect on subsequent trials but does, of course, not preclude overall interference by the additional task. Second, we aimed at reducing the cognitive load of the tRT by increasing the salience of the imperative cues of the tRT (large magenta circles as imperative cues) and by employing different effectors than the main task (i.e., the thumbs instead of index and middle fingers). Finally, by implementing the stimulus-response matching through spatial congruency (i.e., left cue – left thumb, right cue – right thumb), we aimed at further reducing the cognitive load (e.g.,⁴).”

[Supplementary Note 2 Limitations of the Thumb reaction time task]

If the reviewers and/or the editor consider it more appropriate, we would of course agree to move this limitations paragraph into the main manuscript.

In the description of the EEG you say 10-10 system – do you mean 10-20 system as this is the most commonly used EEG – recorded 127 electrodes. Say 10-10 system – do you mean 10-20 or did you use a 10-10 configuration?

We verified the electrode layout with the manufacturer (<https://pressrelease.brainproducts.com/uol/>) and corrected the specifics of the electrode layout now in the manuscript accordingly:

“Continuous EEG was recorded from 127 cephalic active surface electrodes (actiCAP, BrainProducts GmbH, Gilching/Germany) arranged according to the extended international 10-20 system and referenced to the FCz electrode (implicit reference).”

[revised manuscript, line 766ff]

In section “Analysis of the EEG data” you state that you analyse the ISPC between motor-cortical source and the signal from intrinsic hand muscles but everywhere else you simply state that you compare left and right sensorimotor cortex. Did you do any comparison with the hand muscles? If so this should be included somewhere, if it is a mistake then it needs to be removed and rewritten.

Indeed, this is a residual of a former version of the manuscript, which included the cortico-muscular connectivity. For ease of readability, we decided to limit the results to the cortico-cortical connectivity. The methods section was now revised accordingly:

“EEG data were analyzed with the main focus on phase-related connectivity (ISPC) between motor-cortical source.” [revised manuscript, lines 918f]

Results:

Figure 1 really helps to understand the details of this complex analysis

We appreciate the positive feedback.

At some points in the manuscript you refer to primary motor cortex (both for MRS and EEG). With MRS and to a certain extent with EEG it is hard to state with confidence that you are only recording from M1 not S1 so I would alter it to sensorimotor cortex throughout.

We agree that specificity is limited for both methods and have changed the labelling consistently throughout. – All occurrences are highlighted in the manuscript and supplemental material. Respective figures and their legends (Fig. 2, 5, 7, Supplemental Fig. 1, 2) have changed accordingly without highlighting. Considering the number of occurrences, we refrain from listing them here in the rebuttal letter.

I’m not an expert in mediation analysis so not sure if Bayesian moderated mediation analysis is most appropriate but it seems sensible from what you’ve described.

For further justification of our choice to employ a Bayesian model to investigate the multimodal associations, we would like to refer to our response to comment 2 of review 1.

Figure 6 – what does pd stand for? You mention it represents likelihood but can you explain what pd stands for.

We agree that the figure legend requires more information to interpret the measures used for inference. The revised legend of figure 6 reads now:

“Figure 6 Results of Bayesian moderated mediation models. a) Overview over posterior directions (pd) for indirect (mediation) effects on error rate (upper matrix) and transition latency (lower matrix) conditional on upper/lower quintiles of moderator age (depicted as YOUNG and OLDER) for the models estimated with the independent variable (IV) based on the three time × frequency clusters derived in the response-locked ISPC analysis. Models were run separately for left and right S/M1 GABA+ as mediator. The pd can be interpreted as the maximum probability of the effect accounting for the evidence derived from the data. Color coding of pd represents likelihood (in %) and direction of effect, i.e. red shading for positive effects and blue shading for negative effects. A pd of 95, 97.5, 99.5, and 99.95% corresponds to the frequentist 2-sided p-value at the thresholds 0.1°, 0.05*, 0.01, 0.001*** respectively. See respective Materials and Methods section for further information about effect descriptors and measures of uncertainty used in Bayesian statistics.”**

[revised manuscript, legend figure 6]

We also added a reference to the methods section, in which we give a more detailed explanation of the measure and its interpretation in lines 1015ff of the revised manuscript:

“To allow for inferences about the relevance of the effects, probability of direction is reported (pd) for posterior probabilities, which can be interpreted as the probability (expressed in percentage) that a parameter (described by its posterior distribution) is strictly positive or negative when accounting for the evidence obtained from the actual data. The pd can take values between 50 (one half on each side) and 100 (fully on either side) and is approximated to a frequentist 2-sided p-value with the formula $p\text{-value} = 2*(1\text{-pd}/100)^{93,94}$. Hence, a pd of 95, 97.5, 99.5, and 99.95% corresponds to p-value at the thresholds 0.1, 0.05, 0.01, 0.001.”

Discussion:

First paragraph – split sentence starting “using a multimodal approach...” into more, shorter sentences – hard to follow as it is and not concrete enough on findings – “GABA+ concentration to allow for” for example doesn’t seem meaningful as it is – needs to be reworded a little to make clearer.

We followed the reviewer’s suggestion and rephrased the first paragraph of the Discussion. It now reads as follows:

“Flexibly adjusting ongoing behavior and switching between different modes of action is an essential ability in the human behavioral repertoire. Unfortunately, this flexibility declines across cognitive domains with increasing age¹. Therefore, we tested dynamic motor-state transitions as a prototypical behavioral paradigm to investigate the effect of age on the interplay between endogenous GABA and the brain’s responsiveness during flexible behavior. Based on this multimodal approach, we provide converging evidence for age-related differences in the sweet spot of endogenous GABA+ concentration that is optimal for interregional neural communication and that benefits flexible behavior. In conclusion, we tentatively suggest that the nature of the neural and neurochemical findings represents indicators for age-related compensatory mechanisms which serve to alleviate deterioration.”

[revised manuscript lines 500ff]

Reviewer #3 (Remarks to the Author):

This paper examines whether GABA mediates the link between functional connectivity and behaviour, and whether this mediation is moderated by age. This is an important contribution to the field and furthers our understanding of the age effects on associations between neural activity and behaviour.

My main recommendation is to fine tune the writing a bit more so as to help the reader follow the arguments. Given that the methods are complex, it is easy for an unfamiliar reader to lose track of the various nuanced findings.

We thank the reviewer for this constructive comment. In the revised manuscript, we tried to further clarify methods, results, and their interpretation following all reviewers’ very thoughtful comments

and recommendations. All sections indicated by reviewer #3 have been highlighted in the revised manuscript and are discussed in detail below.

Major comments

1. Please discuss the significance of decreasing GABA with higher GM fraction in older adults. Along the same lines, what is the advantage of using tissue-corrected GABA values and also including GM fraction in the statistical models?

It is well accepted that the tissue-composition of the voxel of interest accounts for large variance in the quantification of MRS-derived GABA concentration (Harris et al. (2015). PLOS ONE 10:e0117531 <https://doi.org/10.1371/journal.pone.0117531>), which renders tissue-correction meaningful. The quantification method implemented in the Gannet toolbox (version 3.0) assumes a 2:1 signal ratio for the estimation of the GABA concentration ratio between grey (GM) and white matter (WM). While these assumptions are widely accepted for the young and healthy, it has been criticized because it may not hold across the adult age range (for more in-depth discussion, we would like to refer to Maes et al. (2018) Human Brain Mapping, 39(9), 3652–3662 <https://doi.org/10.1002/hbm.24201>). There are some indications, that age-related functional and structural declines of the motor-cortical GABAergic system are not linear processes and may not parallel the age-related GM atrophy (Rozycka & Liguz-Leczna (2017) Aging Cell, 16(4), 634–643. <https://doi.org/10.1111/accel.12605>). Therefore, we decided to include GM fraction into the statistical model, despite having used a tissue-correction method to directly investigate the effect of varying GM fraction on the GABA+ concentration in the voxels of interest between the two age groups. Our results suggest that while GM atrophy is present in the older as compared to the young participants, the reduction of GABA+ concentration does indeed not show the same trend as the GM atrophy (i.e., decreasing GM is not strictly paralleled by decreasing GABA+ concentration). Several non-linear structural or functional alterations might serve as an explanation for this observed divergence in GM atrophy and MRS-derived GABA+ concentration. Conceivable mechanisms are an altered regulation of GABA synthesis (Marczynski (1998) Brain Research Bulletin, 45(4), 341–379 [https://doi.org/10.1016/S0361-9230\(97\)00347-X](https://doi.org/10.1016/S0361-9230(97)00347-X)), changes in GABA reuptake and transporter activity (Scimemi (2014) Frontiers in Cellular Neuroscience, 8 <https://doi.org/10.3389/fncel.2014.00161>), or GABA receptor availability (as found with PET in older adults, Cuyper et al. (2021) NeuroImage, 226, 117536. <https://doi.org/10.1016/j.neuroimage.2020.117536>). Furthermore, these results question the assumption that a 2:1 signal ratio between GM and WM is stable across the lifespan and requires careful consideration in the application of standard quantification methods in ageing populations.

We added a reference to the supplemental discussion to the main manuscript:

“Relative to the young, the older adults showed overall lower GABA+ levels with increasing GM fraction ($\beta = -0.49 \pm 0.19$, 95%CI [-0.86, -0.12], $X^2 = -2.61$, $p < .01$, GROUP \times raw GM

fraction (centered), Type II Wald $X^2(1) = 6.82$, $p < .01$, Figure 2d) across all voxels (for further discussion of this result see Supplementary Note 1).” [revised manuscript lines 135ff]

The supplemental discussion of this result reads:

“Discussion of decreasing GABA+ concentration with higher GM fraction in the older adults

Including GM fraction into the statistical model in addition to employing a tissue-correction method allowed us to directly investigate the effect of varying GM fraction on the GABA+ concentration in the voxels of interest between the two age groups. Our results suggest that while GM atrophy is present in the older as compared to the young participants, the reduction of GABA+ concentration does not parallel the GM atrophy in the older. While one would intuitively assume decreasing GABA+ levels with reduced GM fraction, age-related structural and functional alterations might not necessarily follow a linear trend. Among others, an initial upregulation of GABA synthesis¹, changes in GABA reuptake and transporter activity², or increases in GABA receptor availability measured with PET³ may go alongside with GM atrophy and lead at least transiently to increased GABA+ concentration as measured with MRS. These results require careful consideration in the application of standard GABA quantification methods in ageing populations.” [Supplemental Note 1]

2. Why did you choose not to report the GABA concentration as a ratio to another compound like Creatine?

We agree that the choice of a suitable reference compound is an essential point when interpreting GABA MRS data. Among other metabolites frequently used as reference compounds for GABA, creatine also changes with age (reviewed in Cleeland et al. (2019) *Neuroscience & Biobehavioral Reviews*, 98, 306–319. <https://doi.org/10.1016/j.neubiorev.2019.01.003>; Chiu et al. (2014). *AGE*, 36(1), 251–264. <https://doi.org/10.1007/s11357-013-9545-8>), which complicates the interpretation of any age-related alterations in GABA concentrations when referenced to Cr. Therefore, we decided to quantify GABA relative to water in agreement with previous work of our group and with the work of other groups focusing on ageing populations (e.g., Cassady et al. (2019) *NeuroImage*, 186, 234–244. <https://doi.org/10.1016/j.neuroimage.2018.11.008>; Chalavi et al. (2018) *Neurobiology of Aging*, 66, 85–96. <https://doi.org/10.1016/j.neurobiolaging.2018.02.014>; Hermans et al. (2018) *Neurobiology of Aging*, 65, 168–177. <https://doi.org/10.1016/j.neurobiolaging.2018.01.023>; Maes et al. (2021) *NeuroImage*, 231, 117871. <https://doi.org/10.1016/j.neuroimage.2021.117871>).

In the main manuscript, we described the correction method as:

“The unsuppressed water signal serving as the reference compound ⁵⁷, was fit with a Gaussian-Lorentzian model.” [revised manuscript, line 711f]

3. Single trial phase angle difference – behaviour difference: what does the association between phase angle difference and error rate look like when participants are not dichotomized based on GABA levels?

We thank the reviewer for this question and we have now moved this information from supplementary results to the main text of the revised manuscript:

“The distribution of phase angle differences between left and right S/M1 sources confirmed non-uniformity, i.e. significant clustering of phase angle differences around 0° for the young in the low beta range (15-22Hz: $z = 70.43$, $p_{FDR} = 1.76e-30$) and for both age groups in the high beta range around 0° for young and around 180° for the older (YOUNG: $z = 4.69$, $p_{FDR} = .03$, OLDER: $z=7.26$, $p_{FDR} = .003$) when pooled over transition conditions (Supplementary Figure 3).”

And we added a new supplementary figure to illustrate these results:

Supplementary Figure 3 Phase angle differences pooled over transition modes within group for low beta (15-22Hz, top row) and high beta (25-30Hz, bottom row) frequency bands. Rose plots show histogram of binned phase angle differences with mean direction (red line) and 95% CI (black circumference) for significant non-uniformity of distribution for older (left) and young group (right). Line plots depict circular correlation of phase angle differences with post transition error [in %] shown separately for age groups (yellow – young, blue – older). Both age groups show lower errors after transition for 0° (or 360°, respectively) phase lag between left and right S/M1 at the time of transition.

[Supplementary Figure 3]

4. Supplementary fig.3 – please discuss this figure further. Especially in older participants, the pattern looks very similar to the findings about phase difference during the transition. While the findings are still interesting, if the same pattern is seen for baseline- and during transition- phase differences, interpretation of the data may need to be more nuanced.

I see that this issue is covered in the discussion – it would be helpful to edit the wording in the results section so it doesn't seem like this observation is being dismissed as insignificant.

We agree that it is important to discuss this point. However, we would like to avoid mixing reporting of results and discussing them because we have the impression that this would further complicate reading the text. Following the reviewer's advice, we changed our wording in the results section to mark this point as relevant and, at the same time, avoid dismissing this result as irrelevant. Of note, due to adding a figure, the numbering of the supplementary figures has changed. Therefore, the respective paragraph refers now to Supplementary Figure 4 instead of Supplementary Figure 3. The paragraph now reads as follows:

“Importantly, this association broadly resembled the pattern during transition described above although it was less specific for the within-age group GABA level in the low beta range (descriptive and inferential statistics in Supplementary Tables 12-14, Supplementary Figure 4).”

[revised manuscript, line 375ff]

5. Lines 457-461: the wording is unclear. Do both young and older adults have stronger associations between connectivity and behaviour in the presence of low GABA, but the associations are in different directions?

We revised this paragraph to improve clarity, it reads now:

“This negative mediating effect has diverging consequences for behavior with respect to the two age groups (Figure 7a). In the young, who showed better performance (i.e. lower error rate) with relatively stronger coupling, this direction of the connectivity-behavior association was more pronounced in the presence of lower non-dominant GABA+ concentration. In the older, who showed the opposite direction of the connectivity-behavior association (i.e., relatively higher error rates with stronger coupling), lower GABA+ concentration pronounced this direction. Higher right S/M1 GABA+ concentration, in contrast, ameliorated the association between stronger coupling and worse performance (i.e. higher error rate) in the older.”

[revised manuscript, lines 463ff]

6. Please discuss the significance of the specific frequency bands that showed significant effects in the mediation analysis. For instance, in the context of previous literature, is there any reason why the low beta range is more important than high beta for this task?

We agree that the discussion of functional relevance of the connectivity within the mu to beta frequency range is of importance for the overall conceptualization of the work. In the revised manuscript, we have now included a more detailed discussion:

“Our unimodal results confirmed previous findings of the relevance of interhemispheric motor-cortical connectivity within the mu to beta frequency ranges for flexible behavior which underlies significant age-effects²². Based on the observation that interhemispheric beta band decoupling was modulated by task complexity (i.e., more pronounced in the more complex transition mode), it may be interpreted as an indicator of inhibitory mechanisms necessary for the coordination of less congruent bimanual movements, as suggested by previous work²²⁻²⁴.”

[revised manuscript lines 515ff]

“We observed a relative decoupling throughout the motor-state transitions for the interhemispheric mu to beta band connectivity in the young. In the older adults, in contrast, we found interhemispheric connectivity to be modulated on level of increased coupling during transitions. This finding of overall increased interhemispheric coupling provides support for the hypothesis of age-related dedifferentiation (e.g.^{8,52}).”

[revised manuscript lines 640ff]

7. Why did you choose to exclude failed transitions in the mediation analysis?

Our main purpose was to evaluate the aging-related differences between the control mechanisms of state transitions. Therefore, we focused on trials in which a transition was successfully executed. The working definition for ‘failed transitions’ (Material and Methods section line 889ff) included transitions that were erroneous throughout the entire time window of interest (figure 1b). Hence, ‘failed transitions’ included trials in which the transition was correctly planned but too slow as well as the completely erroneous trials (i.e., no transition at all). With only 0.3%, the overall odds of completely failed trials were very low. Moreover, we observed a decline in the odds of completely failing over the experiment (Results section, lines 201ff). These results were reassuring for us because they confirmed that the behavioral paradigm was suitable for participants of both age groups. However, since we focused on transition performance, our study design was not sufficiently powered to investigate mechanisms underlying failures.

Nonetheless, we agree that investigating the reasons for and underlying mechanisms of failing to flexibly transition between different motor or cognitive states may be a very exciting question for future work with particular relevance to better understanding aging-related motor-/cognitive decline.

To better support our reasoning for this approach, we added now additional information to the Methods section in the revised manuscript:

“Of note, trials with failed transitions were excluded from all subsequent analysis steps because our main interest was to investigate neural mechanisms underlying successful

flexible transition behavior. Furthermore, the experimental paradigm yielded very low odds of completely failed trials, which did not allow to investigate mechanisms underlying failures in transition maneuvers.” [revised manuscript, lines 892ff]

We additionally refer back to this explanation when describing the mediation analysis:

“Therefore, we made use of Bayesian moderated mediation analysis modelling GABA+ as mediator and age as moderator variable and their impact on the relationship between cortico-cortical connectivity and behavior, i.e. *transition latency* and *error rate* (including *cumulative error rate* and *fully correct transitions* but excluding *failed transition* as described in the statistical analysis of the behavioral parameter error rate).”

[revised manuscript lines 964ff]

8. Please note that having a single GABA measurement, rather than task related GABA is a limitation.

We completely agree that having serial measurements or even a functional measurement of the task-specific modulation of GABA+ concentration would be more informative than a single task-free measurement. We added this limitation to the discussion, which now reads:

“For a complete picture it would be necessary to understand the task-specific modulation of inhibitory mechanisms, which requires the repeated or continuous evaluation of GABA+ concentration. Although informative, a single resting-state measurement, as used in the present study, offers an incomplete picture of the functionality of GABAergic inhibition.”

[revised manuscript, line 649ff]

Minor comments

1. Line 80 – please change the wording. At the moment, it sounds like there were 2 MRS sessions.

We are grateful for bringing our attention to these additional points. We revised/corrected them as outlined below and highlighted them in the revised manuscript/supplemental material:

To clarify the overall experimental flow, we changed the text now as follows:

“The participants underwent in total three sessions, including magnetic resonance spectroscopy (MRS) in the first session and familiarization with the behavioral paradigm (motor-state transitions) in the second session. The third session followed 24 hours after the familiarization and involved electroencephalography (EEG) during task performance.”

[revised manuscript, line 84ff]

2. Line 166 – tap with both index and middle finger simultaneously?

We agree that the behavioral paradigm requires further clarification and added the following information in the revised manuscript

“The control of transitions between motor states was tested with a variation of an established paradigm²⁴⁻²⁷, in which the participants had to rhythmically tap in individually adjusted pace with the index and middle fingers of both hands and to control transitions between two coordinative patterns of different complexity (in-phase/anti-phase, Figure 1a).”

[revised manuscript, line 168ff]

Additionally, we revised the legend of Figure 1a

“The behavioral paradigm involved transitions between a stable (left, mirror-symmetric in-phase tapping of both index or middle fingers synchronously) and a less stable (right, anti-phase tapping, i.e. contralateral index and middle finger synchronously) motor state.”

[revised manuscript, legend Figure 1a]

3. Please make the sub-headings within the mediation analysis results more consistent. The analysis is complex and can be confusing for a reader who is not familiar with the data. For instance, in the sections on lines 454 and 464, it may be easier to use the terms error rate and latency in the headings instead of ‘precision’ and ‘faster’.

Following the reviewer’s suggestion, we changed the subheadings of the mediation results now to more directly reflect the relevant behavioral outcome measures:

“Older adults benefit in precision (i.e., lower error rate) from higher non-dominant GABA+ levels.” [revised manuscript line 461]

“Young adults are faster (i.e., shorter transition latency) with higher connectivity in the presence of lower non-dominant GABA+.” [revised manuscript lines 472f]

4. Please check supplementary figure 1 - the a and b figures look identical, except the white dotted line. Also, the caption says 2a and b.

Thank you for bringing this to our attention. This was indeed a mistake which we unfortunately overlooked. Supplemental Figure 1a should have shown the statistical test results for the cluster-based permutation corrected t-test of the time after visual cue against baseline. We corrected this now and also corrected the figure legend as indicated below.

Supplementary Figure 1 a) *Statistical results of cluster corrected permutation test for significant power changes from baseline, pooled over age groups and transition modes.* Time (in sec) is presented relative to visual cue onset (at 0 sec.). Orange color shading indicates significant power increase and purple color shading indicates significant power decrease with respect to the baseline period [-500 to -200ms relative to cue onset]. Black lines highlight significant frequency-by-time clusters (2-sided t-test, permutation-based cluster correction). **b)** *Statistical results of cluster corrected permutation test for significant power changes for group contrast (OLDER vs. YOUNG) for the transition mode difference (IP - AP).* Orange color shading indicates significant power increase and purple color shading indicates significant power decrease for the OLDER relative to the YOUNG. Black lines highlight significant frequency-by-time clusters (2-sided t-test, permutation-based cluster correction at $p < .05$). Time (in ms) is presented relative to the response (white vertical dashed line at 0ms). Zooming into the time window ± 260 ms around the individual median transition latency for the analysis of the effect of transition mode and its modulation by factor age group, i.e. running a two samples t-test on the contrast $[IP-AP]_{YOUNG} - [IP-AP]_{OLDER}$, revealed specific time and frequency clusters reaching level of significance ($p < .05$, 2-sided) for the individual sources.

[revised Supplementary Figure 1]

5. Line 783 – is it MIMA or MIDA?

It should state MIDA model, which we corrected now:

“For the individual geometrical description of the head (mesh), the anatomical image was segmented into 12 tissue classes (skin, eyes, muscle, fat, spongy bone, compact bone, cortical gray matter, cerebellar gray matter, cortical white matter, cerebellar white matter, cerebrospinal fluid and brain stem), based on the MIDA model⁶⁶ using SPM12 (<http://www.fil.ion.ucl.ac.uk/spm/software/spm12/>), as described previously^{67–69}.”

[revised manuscript, line 804ff]

REVIEWERS' COMMENTS:

Reviewer #1 (Remarks to the Author):

The revision addressed some of my previous concerns (e.g., multiple comparisons, analyzing both left and right hemisphere), but I'm afraid that some of my most serious concerns remain. In particular, I still did not find the work to be well motivated by a specific question, hypothesis or theoretical model. The revision attempts to relate the model to studies of age-related neural dedifferentiation, but it wasn't clear to me how the present results advanced or refuted previous ideas about dedifferentiation. In my previous review I asked if the results disconfirm any plausible hypotheses, and I still have that same question after reading the revision. I did think the authors' response about multiple comparisons was helpful, but the authors themselves concede that much of the work is exploratory rather than being hypothesis driven. Likewise, the authors also concede that behavioral results are limited to variants of a very specific tapping task and so their generalizability is not clear.

Reviewer #2 (Remarks to the Author):

Based on my reviews, I appreciate the inclusion of a paragraph explaining the limitations of the thumb reaction time task and I understand the need to place it in supplementary material in order to keep the manuscript at a reasonable length.

In both the abstract and the Discussion you have altered the text to 'sweet spot' of GABA. I don't think the term sweet spot is particularly clear or useful. Elsewhere you wrote the 'preferred state' of GABA, I wonder if this is what you mean and whether that or 'optimal' would be a better phrase to use.

Apart from this you have addressed my concerns and I am happy to approve this manuscript for publication.

Reviewer #3 (Remarks to the Author):

The authors have adequately addressed my concerns. I have no further comments.

Manuscript COMMSBIO-21-0749A

Title: " The interaction between endogenous GABA, functional connectivity and behavioral flexibility is critically altered with advanced age"

We want to express our gratitude for the appreciation and the valuable comments of all three reviewers. We have carefully considered all their points and followed their suggestions as discussed point by point below. These revisions include additional presentation of data and results as well as major revisions of the text in the main manuscript and the supplementary material to improve clarity of our reasoning, methods, and interpretation of the results.

We feel that this comprehensive revision has improved the quality of our manuscript substantially.

Please find below our detailed responses to each of the reviewers' comments (identified with outside borders with our respective responses directly underneath) and the respective paragraphs highlighted (yellow) in the manuscript and supplements.

Reviewer #2 (Remarks to the Author):

Based on my reviews, I appreciate the inclusion of a paragraph explaining the limitations of the thumb reaction time task and I understand the need to place it in supplementary material in order to keep the manuscript at a reasonable length.

Based on the editorial instructions, we have now included this limitation paragraph in the main manuscript. To keep readability, we have split the paragraph and included the acknowledgment of limitations together with the presentation of the thumb reaction time results with reference to our strategy of minimizing the risk of interference with the main task's performance in the Method section:

“Additional support for comparable transition performance in both age groups comes from the results of the thumb reaction task (methods and results in Supplementary Note 2, Supplementary Table 10), which neither show an effect of group nor interactions with transition mode or time across the experiment. However, it is necessary to acknowledge that the thumb reaction time task poses additional cognitive load throughout the experiment and may, therefore, have interfered with the performance in mode transitions, particularly in the older participants. While we cannot rule out that this interference effect was present and that it has potentially affected the two age groups differently, we took a two-layered strategy to reduce this interference effect as much as possible (see Material section – Behavioral Paradigm).” [revised manuscript p. 9, lines 221ff]

“We followed a three-layered strategy to minimize the risk of interference of the tRT with the performance in the main task: First, we implemented a pause that directly followed each thumb reaction time task. This strategy was intended to reduce the effect on subsequent trials but does, of course, not preclude overall interference by the additional task. Second, we aimed at reducing the cognitive load of the tRT by increasing the salience of the imperative cues of the tRT (large magenta circles as imperative cues) and by employing different effectors than the main task (i.e., the thumbs instead of index and middle fingers). Finally, by implementing the stimulus-response matching through spatial congruency (i.e., left cue – left thumb, right cue – right thumb), we aimed at further reducing the cognitive load (e.g., 68).” [revised manuscript p. 27, lines 659ff]

In both the abstract and the Discussion you have altered the text to ‘sweet spot’ of GABA. I don’t think the term sweet spot is particularly clear or useful. Elsewhere you wrote the ‘preferred state’ of GABA, I wonder if this is what you mean and whether that or ‘optimal’ would be a better phrase to use.

We have revised the two sentences upon the reviewer’s suggestion to improve our statements’ clarity. They read now:

Abstract:

“We provide converging evidence for age-related differences in the preferred state of endogenous GABA concentration for more flexible behavior.” [revised manuscript p. 1, lines 32ff]

Discussion:

“Based on this multimodal approach, we provide converging evidence for age-related differences in the preferred state of endogenous GABA+ concentration that is optimal for interregional neural communication and that benefits flexible behavior.” [revised manuscript p. 17, lines 410ff]